# The CDK-PLK1 axis targets the DNA damage checkpoint sensor protein RAD9 to promote cell proliferation and tolerance to genotoxic stress

Takeshi Wakida[1,2†], Masae Ikura[2], Kenji Kuriya[3], Shinji Ito[4], Yoshiharu Shiroiwa[1‡], Toshiyuki Habu[1,5], Takuo Kawamoto[6], Katsuzumi Okumura[7], Tsuyoshi Ikura[2,8], Kanji Furuya[1,9]*

[1]Department of Radiation Systems, Radiation Biology Center, Kyoto University, Kyoto, Japan; [2]Laboratory of Chromatin Regulatory Network, Department of Mutagenesis, Radiation Biology Center, Kyoto University, Kyoto, Japan; [3]Laboratory of Nutritional Chemistry, Department of Life Sciences, Graduate School of Bioresources, Mie University, Tsu, Japan; [4]Medical Research Support Center, Graduate School of Medicine, Kyoto University, Sakyo-ku, Japan; [5]Department of Food Science and Nutrition, Mukogawa Women's University, Nishinomiya, Japan; [6]Radioisotope Research Center, Kyoto University, Kyoto, Japan; [7]Laboratory of Molecular and Cellular Biology, Department of Life Sciences, Mie University, Tsu, Japan; [8]Laboratory of Chromatin Regulatory Network, Graduate School of Biostudies, Kyoto University, Kyoto, Japan; [9]Laboratory of Genome Maintenance, Graduate School of Biostudies, Kyoto University, Kyoto, Japan

*For correspondence:
kfuruya@house.rbc.kyoto-u.ac.jp

Present address: †Sumitomo Chemical Co., Ltd, Takarazuka, Japan; ‡Aichi Institute of Technology, Toyota, Japan

Competing interests: The authors declare that no competing interests exist.

**Abstract** Genotoxic stress causes proliferating cells to activate the DNA damage checkpoint, to assist DNA damage recovery by slowing cell cycle progression. Thus, to drive proliferation, cells must tolerate DNA damage and suppress the checkpoint response. However, the mechanism underlying this negative regulation of checkpoint activation is still elusive. We show that human *Cyclin-Dependent-Kinases* (CDKs) target the RAD9 subunit of the 9-1-1 checkpoint clamp on Thr292, to modulate DNA damage checkpoint activation. Thr292 phosphorylation on RAD9 creates a binding site for *Polo-Like-Kinase1* (PLK1), which phosphorylates RAD9 on Thr313. These CDK-PLK1-dependent phosphorylations of RAD9 suppress checkpoint activation, therefore maintaining high DNA synthesis rates during DNA replication stress. Our results suggest that CDK locally initiates a PLK1-dependent signaling response that antagonizes the ability of the DNA damage checkpoint to detect DNA damage. These findings provide a mechanism for the suppression of DNA damage checkpoint signaling, to promote cell proliferation under genotoxic stress conditions.
DOI: https://doi.org/10.7554/eLife.29953.001

## Introduction

To proliferate and maintain the integrity of biological systems, cells possess the ability to overcome environmental stresses. Genomic DNA damage is a type of environmental stress, and when cells are exposed to genotoxic stress, cell cycle progression is delayed by a mechanism called the DNA damage checkpoint. Although the delay increases the time available for DNA repair processes to perform damage recovery (*Elledge, 1996*; *Weinert and Hartwell, 1988*), the degree of the checkpoint response must be tightly regulated, depending on the proliferative status of the cells. Notably, if

cells must proliferate, then the checkpoint response should be down-regulated to tolerate the cellular DNA damage stresses. This may be accomplished via the interplay between the pro-mitotic activity and the DNA damage checkpoint in the cell, although the precise mechanism remains enigmatic (*Serrano and D'Amours, 2014*).

Among the pro-mitotic activities, CDKs are the primary driving forces that order and coordinate cell cycle events (*Swaffer et al., 2016*). Activation by CDKs not only initiates M-phase events, but also controls temporally ordered programs throughout the cell cycle, such as replication origin firing upon initiation of DNA replication, which potentially support proliferation (*Stern and Nurse, 1996*; *Tanaka et al., 2007*; *Trovesi et al., 2013*; *Zegerman and Diffley, 2007*). However, problems may arise once DNA replication is initiated. The DNA replication forks can encounter damaged DNA or environmental stress-mediated nucleotide depletion, and such events can stall replication polymerases at the fork (*Jossen and Bermejo, 2013*). The stalled replication forks expose single-stranded DNA, which is recognized by checkpoint signaling complexes that activate the S-phase checkpoint (*Giannattasio and Branzei, 2017*; *Katou et al., 2003*; *Sogo et al., 2002*; *Zou and Elledge, 2003*; *Zou et al., 2003*). When the DNA damage response is dominant, checkpoint signaling reduces the rate of S-phase progression by repressing late origin firing or by slowing the progression of individual replication forks. Together, these events can reduce genomic instability in proliferating cells, and thereby provide tumor-suppressor functions (*Iyer and Rhind, 2013*; *Zegerman and Diffley, 2009*).

Interestingly, under some circumstances, pro-mitotic activity dominates DNA damage checkpoint signaling, and cells commence proliferation even in the presence of damaged DNA. The key protein involved in the pro-mitotic activity to override DNA damage checkpoint signaling is PLK1. This phenomenon was initially reported as *adaptation*, and was originally described in a budding yeast system in which cells overrode a G2 DNA damage checkpoint via a CDC5 (budding yeast PLK1 orthologue)-dependent mechanism, after prolonged G2 arrest induced by irreparable DNA double-strand breaks (DSBs) (*Pellicioli et al., 2001*; *Sandell and Zakian, 1993*; *Toczyski et al., 1997*). *Adaptation* was also reported in a *Xenopus* system, in which a prolonged aphidicolin-induced replication block was compromised by a Plx1 (Xenopus PLK1 orthologue)-dependent process (*Yoo et al., 2004*), and an analogous phenomenon has also been reported in mammalian cells. However, mammalian cells only activate the G2/M DNA damage checkpoint when a certain amount of DNA damage (e.g. ~20 DSBs) is present. Indeed, mammalian cells can enter mitosis even in the presence of DNA damage signals, such as γ-H2AX foci (*Deckbar et al., 2007*; *Ishikawa et al., 2010*; *Syljuåsen et al., 2006*). In this sense, mammalian cells can somehow repress a DNA damage checkpoint mechanism to tolerate the DNA damage response in order to drive proliferation, and PLK1 exhibits the key pro-mitotic activity for this purpose. When the cellular PLK1 activity reaches a certain level, the cells can re-enter mitosis upon recovery from G2 checkpoint arrest (*Liang et al., 2014*). However, in the case of the DNA damage checkpoint in S-phase, the crosstalk between PLK1 and the DNA damage response becomes more complicated. In fact, PLK1 functions not only upon mitotic commitment, but also during S phase or a related DNA damage response, thereby facilitating DNA metabolism in support of rapid cell proliferation (*Moudry et al., 2016*; *Yata et al., 2012*). Importantly, the PLK1 protein contains a *Polo-Box Domain* (PBD) that recognizes a pre-phosphorylated substrate (*Elia et al., 2003*). The substrate is phosphorylated by a kinase, typically a CDK, and the PBD preferentially recognizes phospho-serine/threonine (-S-pS/pT-P-)-containing peptides (*van Vugt et al., 2010*).

In this study, to clarify how proliferating cells override the DNA damage checkpoint in S-phase, we revisited the CDK-dependent phosphorylation in the checkpoint processes, using human cells. We found that CDK targeted RAD9, a subunit of the DNA clamp complex 9-1-1, which acts as a DNA damage sensor protein complex in checkpoint signaling (*Delacroix et al., 2007*; *Furuya et al., 2004*; *Kumagai et al., 2004*; *Lee et al., 2007*; *Liu et al., 2012*; *Navadgi-Patil and Burgers, 2009*; *St Onge et al., 2003*; *Takeishi et al., 2010*). If CDK failed to phosphorylate RAD9, then the cells accumulated in S phase in the presence of low doses of hydroxyurea (HU). The CDK-dependent phosphorylation of RAD9 at residue 292 (pThr292) created a binding site for PLK1, and the bound PLK1 was then activated to further phosphorylate RAD9 on Thr313 and Ser326. We also found that these CDK- and PLK1-dependent phosphorylations of RAD9 were required to maintain the rate of S-phase progression in cells exposed to low-dose HU, and this correlated with the mutant phenotype in which increased amounts of active checkpoint complexes accumulated on the damaged chromatin. Additionally, mutant RAD9 proteins that could not be phosphorylated by CDK-PLK1 were

also associated with the damaged chromatin. These results implied that the CDK-/PLK1-dependent control of DNA damage detection by RAD9 plays a crucial role in minimizing the DNA damage checkpoint response, thus helping to drive proliferation under conditions of DNA replicative stress.

## Results

### CDK-dependent phosphorylation of RAD9 at Thr292

To understand how CDK controls the DNA damage checkpoint, we focused on the human checkpoint protein RAD9, which forms a PCNA-like hetero-trimeric checkpoint clamp complex. This complex (designated as 9-1-1) functions as a DNA damage sensor by binding to damaged DNA ends (*Caspari et al., 2000*; *Majka et al., 2006*; *Thelen et al., 1999*), and this function must be precisely regulated to control the degree of checkpoint activation. Indeed, using the fission yeast (*Schizosaccharomyces pombe*) system, we previously identified multiple phosphorylation events on the *S. pombe* Rad9 homologue (spRad9) that regulate the checkpoint activation and the spRad9 release from damaged chromatin (*Furuya et al., 2010*, *2004*). In the present study, we focused on Thr292 (-His-Ser-$^{292}$Thr-Pro-) of the human RAD9 homologue (*Figure 1A*), because the residue resembles Thr321 (-His-Ser-Ser-$^{321}$Thr-Pro-) of SpRad9, which when phosphorylated promotes the release of SpRad9 from DNA damage sites.

The phosphorylation of RAD9 Thr292 was originally identified by Davey's group (*St Onge et al., 2003*). As this Thr residue is followed by a proline residue and the region resembles a putative CDK phosphorylation site, we first tested whether the phosphorylation of Thr292 depended on CDK. RAD9 with a C-terminal myc-His (mH) tag was transiently expressed in HEK293A cells, and the cell lysates were subjected to fractionation by centrifugation in sucrose-containing buffer (*Figure 1—figure supplement 1A*)(*Ohashi et al., 2014*; *Zou et al., 2002*). Western blotting was performed using an antibody that recognizes phosphorylated Thr292 (pThr292). We detected the pThr292 signal primarily in the supernatant fraction (*Figure 1—figure supplement 1A*). The pThr292 band corresponded to the uppermost band of the RAD9-myc signal, as previously reported (*Figure 1—figure supplement 1A*) (*St Onge et al., 2003*). The pThr292 signal was diminished in roscovitine (CDK inhibitor)-treated cells, although the band was still detectable in RO-3306 (CDK1 inhibitor)-treated cells (*Meijer et al., 1997*; *Vassilev et al., 2006*). These results suggested that any CDK would be able to phosphorylate Thr292 of RAD9. Importantly, the pThr292 signal was diminished when a mutant RAD9, with Thr292 substituted with alanine (T292A), was expressed in place of wild-type RAD9 (*Figure 1—figure supplement 1A*).

We further tested the ability of CDK to phosphorylate RAD9 Thr292 in an in vitro kinase assay, using a recombinant CDK2-cyclinA2 complex. The C-terminal region of RAD9 (a.a. 266–391) was fused to GST, purified from *E. coli* cells by glutathione affinity chromatography, and assayed to detect CDK-dependent phosphorylation in vitro. pThr292 was detected as efficiently as pSer277 by western blotting (*Figure 1B*) (*St Onge et al., 2003*). We also confirmed that the phosphorylation at Thr292 was not dependent on other CDK phosphorylation sites (Ser277, Ser328, Ser336) that are responsible for the major bandshift of GST-RAD9 in vitro (*St Onge et al., 2003*) (data not shown).

Next, to assess the behavior and effects of the CDK-dependent phosphorylation of RAD9 in vivo, we constructed stable HEK293A cell lines that harbor a *CMV-tetO-RAD9-mH* construct (genomically integrated at the *FRT* locus) and therefore express wild-type or Thr292-mutated (T292A) RAD9-mH. For the in vivo cell line experiment, we used RAD9-S291A/T292A in place of RAD9-T292A, and these two mutant proteins were treated similarly throughout the manuscript. RAD9-mH was expressed when doxycycline was added to the medium (*Figure 1—figure supplement 1B*). Although RAD9-mH was expressed at a level approximately five times higher than endogenous RAD9, we assumed that this increased level of ectopic expression did not affect the protein's usual cellular activity. Extra copies of RAD9 reportedly associate with, and are possibly sequestered by, the CAD (carbamoyl-phosphate synthetase) protein, which does not associate with RAD1-HUS1 (*Lindsey-Boltz et al., 2004*), and thus the expression of the mutant RAD9 would replace the endogenous RAD9 as a component of the 9-1-1 complexes.

A thymidine block and release was performed to synchronize the cells in G1/S, and the cell cycle profile of pThr292 was monitored. After the cells were released from the thymidine-induced G1/S block, the cells that expressed RAD9-mH from either *CMV-tetO-RAD9-mH* or *CMV-tetO-RAD9-*

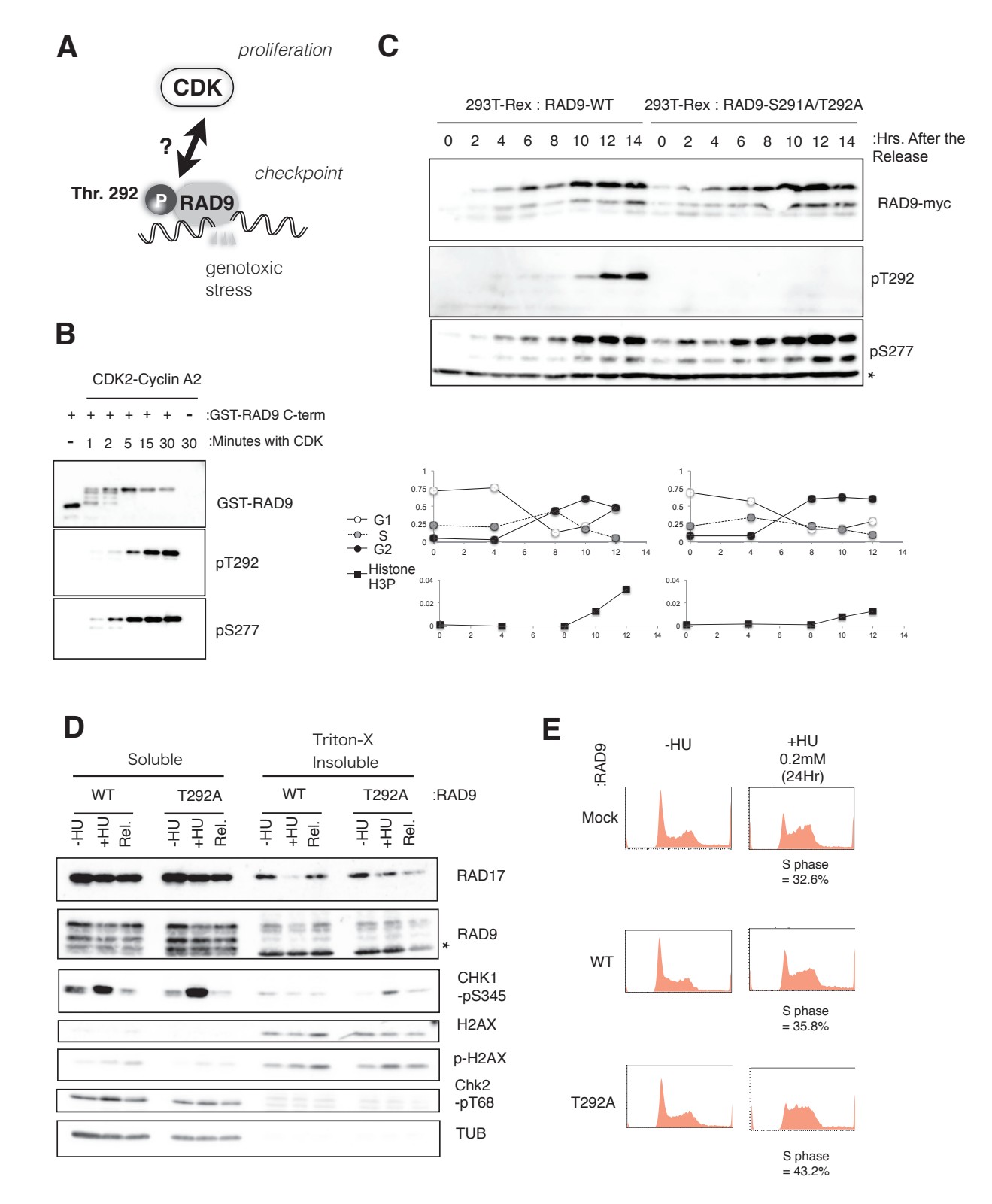

**Figure 1.** CDK phosphorylates threonine 292 of RAD9. (**A**) Schematic of the aim of this manuscript. (**B**) The recombinant GST-tagged C-terminal (a.a. 266–391) portion of RAD9 was mixed with the purified active CDK2-CyclinA2 complex. Western blotting was performed using the α-RAD9 antibody and the α-pT292 (pT292) and α-phospho-Ser277 (pS277) RAD9 antibodies. (**C**) *Top*: The thymidine block and release experiment was performed on HEK293A-T-REx cell lines stably expressing RAD9-WT-mH or RAD9-S291A/T292A-mH. Doxyclin (0.5 μg/ml) was added during the second thymidine

*Figure 1 continued on next page*

*Figure 1 continued*

block. The western blotting analysis of cell lysates obtained at the indicated time points is shown, and α-myc, α-pT292 and α-pS277 antibodies were used. The asterisk (*) shows a nonspecific signal. *Bottom*: The cell cycle profiles were quantified by a flow cytometer analysis. The G1, S, and G2 phases were quantified via propidium iodide staining, and the M phase cells were quantified using an antibody against the phospho-serine 10 of histone H3. (D) A chromatin fractionation assay (*Ohashi et al., 2014*; *Zou et al., 2002*) was performed in HEK293A-T-REx cell lines stably expressing RAD9-WT-mH (WT) or RAD9-S291A/T292A-mH (T292A). Cells were grown in media containing 1.5 mM hydroxyurea for 16 hr (+HU) and then released from the hydroxyurea arrest for 1 hr (Rel.). A western blotting analysis is shown, and α-RAD17, α-myc (RAD9), α-phospho-serine 345 of CHK1 (CHK1-pS345), α-phospho-serine 139 of Histone H2AX (p-H2AX), α-H2AX, α-phospho-threonine 68 of CHK2 (CHK2-pT68), and α-tubulin were used. (E) The flow cytometry analysis was performed with U2OS T-REx cells stably expressing RAD9-WT-mH (WT) or RAD9-S291A/T292A-mH (T292A), and the host U2OS T-REx cells (Mock). The cells were grown in media containing 0.2 mM hydroxyurea for 24 hr. The populations of cells showing the S phase peaks were quantified and indicated below the flow cytometer profiles. See also *Figure 1—figure supplement 1*. CDK phosphorylates threonine 292, and construction of RAD9-WT or -S291A/T292A(T292A) expressing stable cell lines.
DOI: https://doi.org/10.7554/eLife.29953.002

The following figure supplement is available for figure 1:

**Figure supplement 1.** CDK phosphorylates threonine 292, and construction of the RAD9-WT and -S291A/T292A(T292A) expressing stable cell lines.
DOI: https://doi.org/10.7554/eLife.29953.003

*S291A/T292A-mH* were harvested and subjected to a western blotting analysis, using anti-pThr292 (pT292: *Figure 1C*). The phosphorylation of Thr292 was observed weakly from mid S phase (6–8 hr in *Figure 1C*) and strongly at the G2/M transition, a pattern that correlated with the CDK activity. In combination with the above results, these experiments confirmed that CDK phosphorylates the Thr292 residue of RAD9 both in vitro and in vivo.

## Checkpoint signaling is enhanced upon RAD9-T292A expression

We observed a slight delay in the mitotic commitment, as detected by the delayed appearance of the phosphorylation of histone H3 Ser10, in cells expressing T292A-mutated RAD9 (*Figure 1C*, bottom). This implied a role of the CDK-dependent phosphorylation (pThr 292) in inhibiting the RAD9 activity for the DNA damage checkpoint.

Next, we sought to determine how the CDK-dependent phosphorylation of RAD9 affected the checkpoint-activating processes at the molecular level. To do so, we monitored the amount of chromatin-bound checkpoint proteins in HEK293A-T-REx cells expressing either wild-type or T292A-mutated RAD9 under replicative stress (1.5 mM hydroxyurea), by performing a fractionation assay followed by a western blotting analysis (*Figure 1D*) (*Ohashi et al., 2014*; *Zou et al., 2002*). Tubulin and H2AX were specifically detected in the Triton X-100 soluble (S1) and insoluble (P1/P2) fractions, respectively, suggesting that the fractionation functioned appropriately. The majority of the RAD9-myc was found in the S1 fraction, as previously reported, and approximately 10% of the RAD9-myc was found in the insoluble fraction. The proportion of wild-type RAD9 (WT) in the insoluble fraction decreased after the cells were exposed to HU (see '+HU' in *Figure 1D*). In contrast, the T292A-mutated RAD9 (T292A) remained in the insoluble fraction when the cells were exposed to HU. A previous report showed that upon the activation of the checkpoint, the 9-1-1 complex is loaded onto DNA damage sites by the RAD17 clamp loader complex and the DNA damage signal is transmitted to the CHK1 kinase (*Zou et al., 2002*). Indeed, higher levels of RAD17 and the Ser345-phosphorylated CHK1 accumulated in the insoluble fraction of HU-treated cells expressing the T292A-mutated RAD9 (+HU, compare WT and T292A in *Figure 1D*). The increased amounts of checkpoint components in the chromatin fraction suggested that the checkpoint is enhanced in cells expressing the T292A-mutated RAD9.

## S-phase progression is slowed in cells exposed to low-dose HU when CDK fails to phosphorylate RAD9 Thr292

The enhanced association of RAD9 and phosphorylated CHK1 with chromatin in the presence of HU suggested the ectopic activation of the intra-S-phase checkpoint, when CDK failed to phosphorylate RAD9 at Thr292. The intra-S-phase checkpoint is activated when cells are under dNTP-depletion stress or when replication forks encounter DNA damage (*Paulovich and Hartwell, 1995*). The activation of this checkpoint delays the timing of late origin firing, which is a CDK-dependent process

(*Dimitrova and Gilbert, 2000*; *Santocanale and Diffley, 1998*; *Shirahige et al., 1998*; *Tanaka and Araki, 2010*; *Yekezare et al., 2013*).

To detect S-phase progression in cells, we used U2OS-T-REx cells stably expressing wild-type RAD9 or T292A-mutated RAD9 from the *FRT* locus (*Figure 1—figure supplement 1C*). We performed these experiments in the presence of 0.2 mM HU, a lower concentration of HU that allowed the control U2OS cells to grow with a mild S-phase delay. Here, we assessed the cell cycle delay by flow cytometry, to measure the cellular DNA content (*Figure 1E*). In the absence of HU, the U2OS cells that expressed wild-type RAD9 (WT) and those that expressed the T292A-mutated RAD9 (T292A) behaved similarly to the parental U2OS-T-REx cells. However, in presence of 0.2 mM hydroxyurea, although the cells expressing wild-type RAD9 showed similar profiles to the parental U2OS cells, the cells expressing the T292A-mutated RAD9 exhibited an increased population of S-phase cells (*Figure 1E*, WT: 35.8%, T292A: 43.2%).

The accumulation of S-phase cells could be due to a slower rate of DNA synthesis. Therefore, we quantified the rate of dNTP incorporation by EdU (ethynyl-deoxy-uridine)-labeling (*Salic and Mitchison, 2008*). Without HU treatment, both cell lines (expressing wild-type or T292A-mutated RAD9) showed similar amounts of EdU incorporation during S phase. However, when these cell lines were treated with 0.2 mM HU for 24 hr, the amount of EdU incorporation in the cells expressing the T292A-mutated RAD9 was reduced to 60–70% of that in the HU-exposed cells expressing wild-type RAD9 (details shown later in Figure 4B, compare WT and T292A). This result suggested that the higher rate of genomic DNA replication under dNTP depletion stress was maintained through the CDK-dependent phosphorylation of Thr292 of RAD9.

## PLK1 binds to, and is activated by, the phosphorylated Thr292 of RAD9

Next, we explored the biochemical consequences of the CDK-dependent phosphorylation of Thr292 (pThr292) of RAD9. We noticed that the peptide sequence around pThr292 (289-Ala-His-Ser-*pThr*-Pro-293) resembles the cognate sequence for PLK1 binding (*Elia et al., 2003*). PLK1 is a kinase with a C-terminal phospho-peptide-recognizing motif called the PBD, which is known to bind to a Ser-pSer/pThr-Pro consensus motif. Indeed, we found that the PBD of PLK1, but not those of PLKs2-5, efficiently bound to the C-terminus of RAD9 in a yeast two-hybrid assay (*Figure 2—figure supplement 1A*). We further demonstrated that this binding decreased when Thr292 of RAD9 was substituted with alanine (*Figure 2—figure supplement 1A*).

We then performed an in vitro 'pull-down' experiment using recombinant GST-RAD9 protein (C-terminus: a.a. 266–391) and PLK1. In the course of the assay, the GST-RAD9 protein was phosphorylated by a recombinant CDK2-cyclinA2 complex. PLK1 co-precipitated efficiently using GSH-beads, but only when CDK2-cyclinA2 was present (*Figure 2A and B*). Notably, this experiment employed a kinase buffer containing ATP, and we observed a bandshift of GST-RAD9 only when PLK1 was present (lane three in *Figure 2A*, lane one in *Figure 2B*). Consistent with the notion that PLK1 binds to and phosphorylates pre-phosphorylated substrates, we observed enriched Thr292 phosphorylation (pT292) in the bandshifted form of GST-RAD9 (*Figure 2A*). We should also note that hereafter, we used GST-RAD9 in which S277, S328 and S336 were each substituted with alanine; the resulting GST-RAD9 (3A) construct facilitated the recognition of PLK1-induced phosphorylation by the bandshift (*Figure 2—figure supplement 1B*).

If PLK1 binds to Thr292-phosphorylated RAD9, then the binding should be competed with a phospho-peptide that contains pThr292. Therefore, we incubated PLK1 and GST-RAD9 together with various phospho-peptides, and monitored the association of PLK1 with the GSH-bead fraction. PLK1 remained in the supernatant fraction when incubated with a pThr292-containing peptide (pT292), but not when incubated with other peptides (Non-P, pS291, pS291/pT292) or in the absence of a competing peptide (*Figure 2C*; compare Sup and Beads).

To further confirm the specific interaction between pThr292 and PLK1, we tested whether a pThr292-containing peptide could activate PLK1. The PBD can reportedly act as an auto-inhibitor of the PLK1 kinase domain, and the binding of a substrate to the PBD counteracts this auto-inhibitory function (*Jang et al., 2002*). Here, we used GST-RAD9 (3A)-T292A (which PLK1 cannot bind) as the substrate for kinase assays using PLK1 (*Figure 2D*). The bandshift in the anti-RAD9 western blot was only observed in the presence of a pThr292-containing peptide, and the use of other peptides in the reaction did not yield a bandshift (*Figure 2D*). These data suggested that a pThr292-containing phospho-peptide binds to and 'activates' PLK1, to promote the further phosphorylation of RAD9.

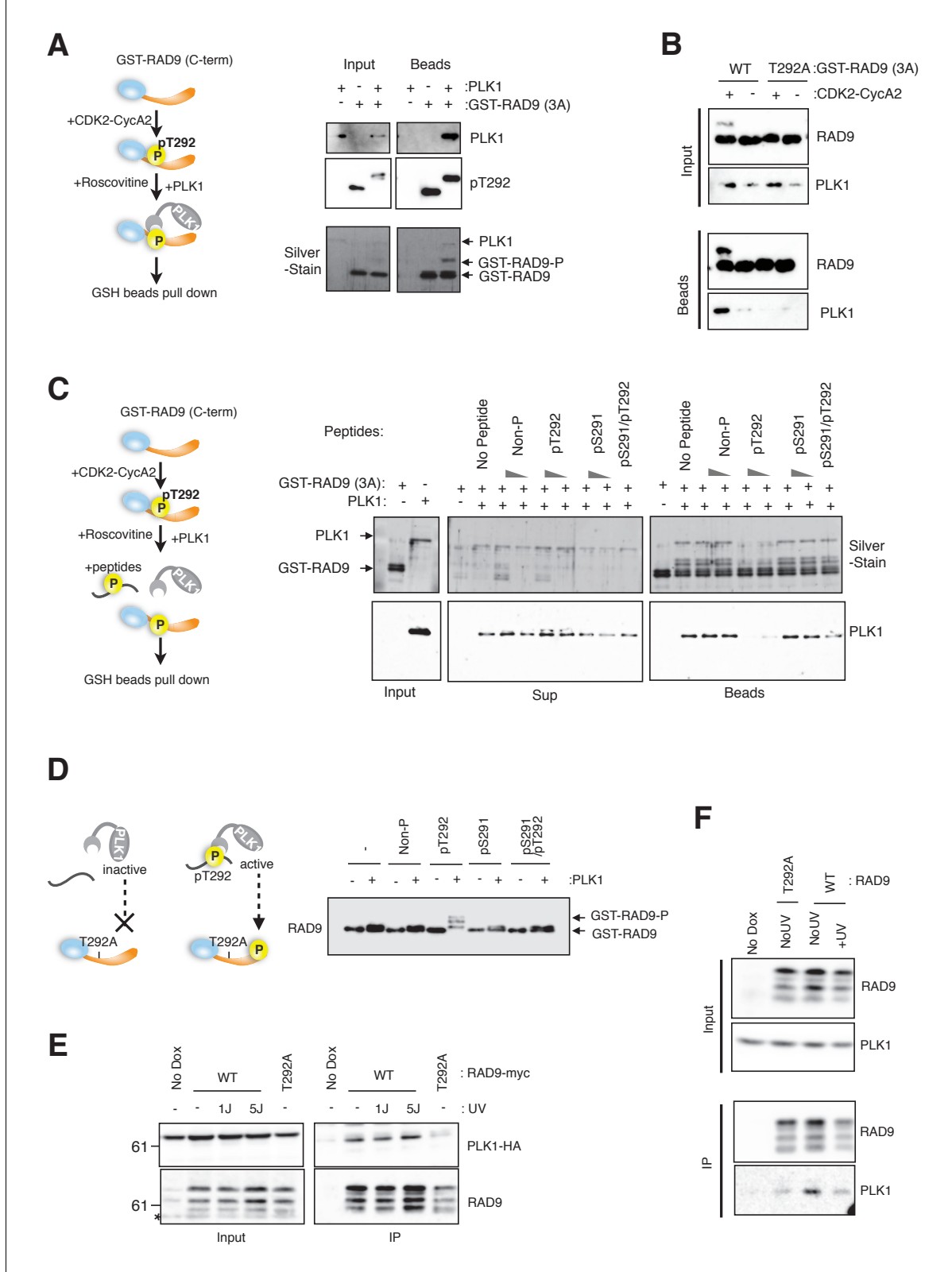

**Figure 2.** CDK-dependent phosphorylation of threonine 292 of RAD9 accommodates and activates PLK1. (**A**) The RAD9 C-terminus co-precipitates with PLK1. GST-RAD9 (3A: S277A, S328A, S336G) (30 pmol) was pre-incubated with CDK2-Cyclin A2 (0.5 pmol) in 15 μl of kinase buffer at 30 ˚C for 30 min, followed by an incubation with 4 pmol of PLK1 (30 ˚C for 5 min followed by 4 ˚C for 30 min). The reaction mixture was captured with GSH-beads for 30 min at 4 ˚C. A schematic of the experiment is shown (*left*). A western blot analysis was performed using α-PLK1 (PLK1) and α-pT292 (pT292) antibodies,
*Figure 2 continued on next page*

Figure 2 continued

and a silver-stained gel is also shown (*right*). (**B**) CDK and Thr292 of RAD9 are responsible for the PLK1-RAD9 interaction. A GSH-bead pull down assay was performed as described in (**A**), except that the reactions without CDK2-CyclinA2 or GST-RAD9-T292A (T292A) mutant protein were added to the experiment. A western blot analysis was performed using α-PLK1 and α-RAD9 antibodies. (**C**) The Thr292-phosphorylated peptide can compete with the RAD9-PLK1 interaction. PLK1 (4 pmol) was pre-incubated with phospho- or non-phospho-peptides (8 nmol or 0.8 nmol, Non-P: non-phosphorylated peptide, pT292: a peptide phosphorylated on Thr292, pS291: a peptide phosphorylated on Ser291, pS291/pT292: a peptide phosphorylated on both Ser291 and Thr292; for the pS291/pT292 peptide, only 8 nmol was tested) for 30 min at 4 °C, and then mixed with GST-RAD9A (3A) (30 pmol), which was phosphorylated by CDK2-CyclinA2 (0.5 pmol) for 30 min at 30 °C in a 20 µl reaction. The reaction mixture was incubated with GSH-beads for 20 min at 4 °C. A schematic drawing of the experiment is shown (*left*). A western blot analysis was performed using α-PLK1, and a silver stained gel is also shown (*right*). (**D**) The T292 phosphorylated peptide can promote PLK1-dependent phosphorylation of RAD9. GST-RAD9-T292A (3A) (10 pmol) was incubated with PLK1 (0.3 pmol) and non-phospho- or phospho-peptides (0.8 nmol: same peptides used in (**C**)), in a 20 µl reaction for 20 min at 30 °C. A schematic drawing of the experiment is shown (*left*). A western blot using the α-RAD9 antibody is also shown (*right*). (**E**), (**F**) RAD9 co-immunoprecipitates with PLK1. Lysates prepared from 293A-T-REx cells stably expressing WT or T292A-mutated RAD9-mH (RAD9-S391A/T292A expressing cells were used for T292A) were subjected to immunoprecipitation, using α-myc antibody-coated agarose beads. Western blots were performed using α-RAD9 and α-PLK1 antibodies on the input (Input) and immunoprecipitates (IP). Ectopically expressed PLK1-HA was used in (**E**) and endogenously expressed PLK1 was detected in (**F**). See also *Figure 2—figure supplement 1*. PLK1 interacts with and phosphorylates RAD9.

DOI: https://doi.org/10.7554/eLife.29953.004
The following figure supplement is available for figure 2:

**Figure supplement 1.** PLK1 interacts with and phosphorylates RAD9.
DOI: https://doi.org/10.7554/eLife.29953.005

The association of PLK1 with RAD9 was confirmed by the cell line experiments. Using anti-myc-coated agarose beads, co-immunoprecipitation (co-IP) experiments were performed using cell lysates from the RAD9-mH-expressing HEK293A-T-REx cell line. First, we tested the co-IP of transiently-expressed PLK1-HA (*Figure 2E*). The co-IP of PLK1-HA was observed regardless of UV damage stress in cells that expressed wild-type RAD9-mH, and was greatly diminished in cells that expressed the T292A-mutated RAD9-mH (*Figure 2E*). We further confirmed the co-IP of endogenous PLK1. The co-IP of PLK1 was also observed in cells that expressed wild-type RAD9-mH, but was decreased in cells that expressed the T292A-mutated RAD9-mH (*Figure 2F*). Together, these results confirmed that PLK1 specifically targets RAD9 phosphorylated (at Thr292) by CDK, and this targeting was demonstrated to occur in vitro and possibly in vivo. In summary, the CDK-dependent phosphorylation of Thr292 of RAD9 is expected to provide a docking site on RAD9, and to act as a local activator for the further phosphorylation of RAD9 by PLK1.

## PLK1 phosphorylates RAD9 at Thr313 and Ser326

Having established the conditions under which PLK1, but not PLK1-KR (kinase-deficient), phosphorylates CDK-pretreated RAD9 (*Figure 3A*), we sought to identify the RAD9 sites that are phosphorylated by PLK1. The phosphorylated bands of RAD9 were subjected to an LC-MS/MS analysis. As shown in *Figure 3B*, significant amounts of pThr313 and pSer326 were detected in GST-RAD9, when it was incubated with both CDK2-CycA2 and PLK1. pThr312 was relatively weakly detected, and its increase upon a co-incubation with PLK1 was not as dramatic as those observed for pThr313 and pSer326. Furthermore, the Thr313 and Ser326 residues of RAD9 are followed by hydrophobic amino acids, and thus Thr313 and Ser326 match the previously reported PLK1 phosphorylation consensus sequence (*Nakajima et al., 2003*). These results indicated that RAD9 Thr313 and Ser326 constitute sites that are phosphorylated by PLK1.

The region around the Thr313 phosphorylation site is conserved among the vertebrate RAD9A proteins (In this manuscript, RAD9 is used synonymously with RAD9A unless otherwise stated.) (*Figure 3—figure supplement 1A*). The RAD9A proteins are thought to be the mitotic paralogues of RAD9 that diverged from an ancestral Rad9. Interestingly, the RAD9B proteins, which are paralogues that are thought to function in testis and/or to have distinct functions (*Dufault et al., 2003*; *Leloup et al., 2010*; *Pérez-Castro and Freire, 2012*), lack the PLK1 phosphorylation sites identified here (*Figure 3—figure supplement 1B*). These observations suggested that the interaction with CDK-PLK1 may provide the critical difference in the regulation between these RAD9 paralogues.

To further confirm the PLK1-dependent phosphorylation, we raised antibodies against pThr313- and pSer326-containing peptides (*Figure 3—figure supplement 1C*). These antibodies, α-pThr313

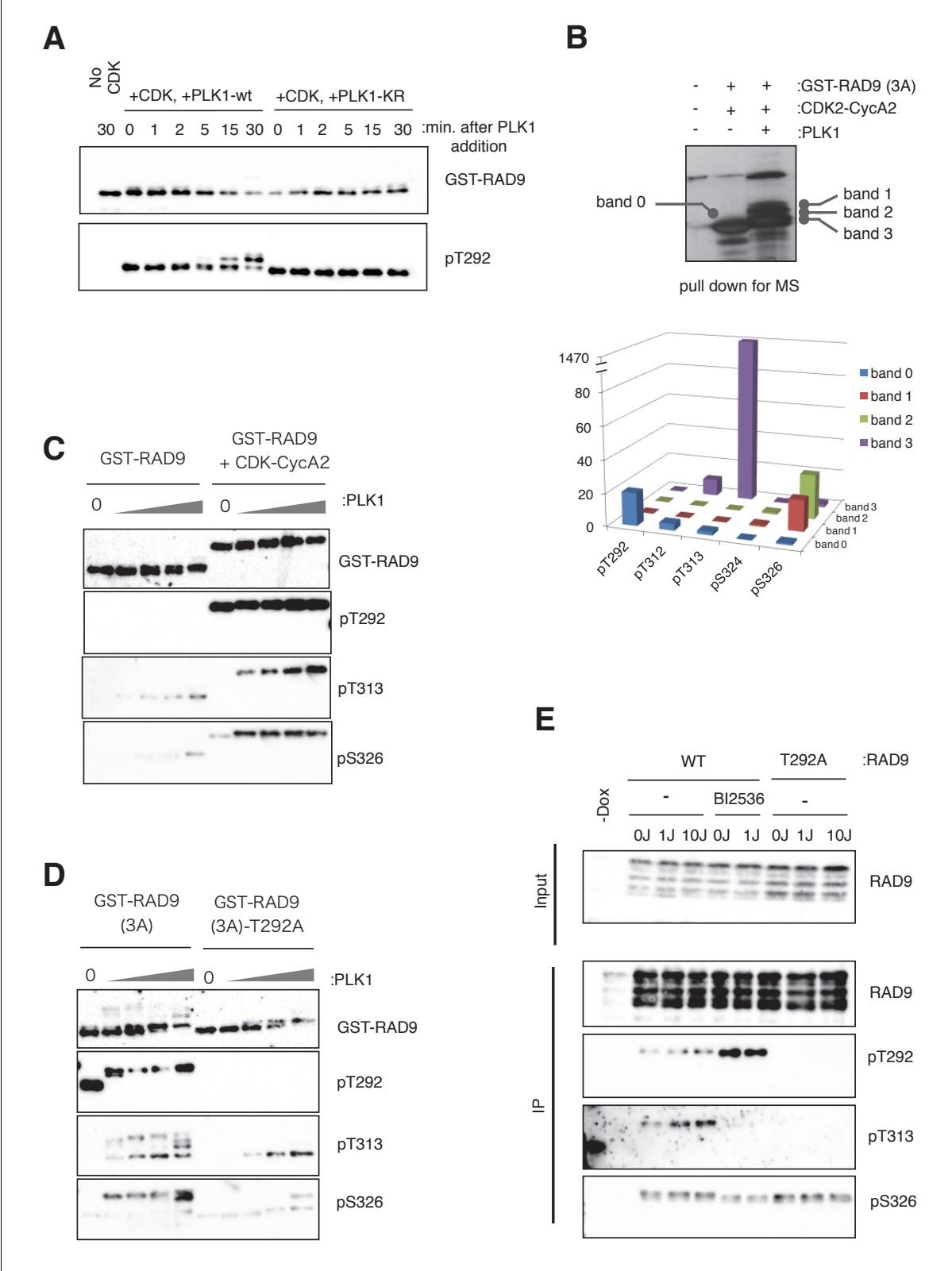

**Figure 3.** PLK1 phosphorylates RAD9 on Thr 313 and Ser 326 in vivo and in vitro. (**A**) PLK1 phosphorylates CDK-phosphorylated RAD9. PLK1 or PLK1-KR (K82R) (0.3 pmol) was incubated with GST-RAD9 (3A) (10 pmol; phosphorylated with the CDK complex (0.5 pmol) for 30 min at 30 °C), at 30 °C for the indicated time points in the presence of roscovitine (10 μM). Samples were obtained and western blot analyses using α-RAD9 (GST-RAD9) and α-pT292 (pT292) antibodies were performed. (**B**) *Top:* Samples for mass spectrometry were prepared by an in vitro CDK-PLK1 kinase assay. GST-RAD9 (3A) (5 μg)
*Figure 3 continued on next page*

Figure 3 continued

was incubated with CDK2-Cyclin A2 (0.5 µg), followed by an incubation with PLK1 (5 µg) in a 100 µl reaction. Phosphorylated GST-RAD9 was pulled down using GSH-agarose beads. A 10% portion of the captured materials was subjected to SDS-PAGE and silver staining. *Bottom:* Mass spectrometry analysis of the PLK1-phosphorylated GST-RAD9 (3A), excised from the bands corresponding to band 0 to band 3 at the *top*. The total counts of the reliable MS/MS spectra (confidence ≥95%) corresponding to the peptides originating from GST-RAD9 (3A), in which Thr292 (pT292), Thr312 (pT312), Thr313 (pT313), Ser324 (pS324), or Ser326 (pS326), was phosphorylated. (C), (D) PLK1 phosphorylates Thr313 (pT313) and Ser326 (pS326) of RAD9, when RAD9 was pre-phosphorylated by CDK on Thr 292 (pT292). GST-RAD9 (10 pmol) was incubated with or without CDK2-CyclinA2 (0.5 pmol) prior to incubations with different amounts of PLK1 (0, 0.3, 0.8, 2.6, 8 pmol). The western blot is shown in (C). GST-RAD9 (3A) (10 pmol) or GST-RAD9 (3A)-T292A (10 pmol) was incubated with different amounts of PLK1 (0, 0.3, 0.8, 2.6, 8 pmol). The western blot is shown in (D). α-RAD9 (GST-RAD9), α-pT292 (pT292), α-pT313 (pT313), and α-pS326 (pS326) antibodies were used for the western blot analyses in (C) and (D). When the GST-RAD9 was incubated with PLK1, roscovitine (10 µM) was added to inhibit the remaining CDK. (E) 293A T-REx cells stably expressing RAD9-WT-mH (WT) or RAD9-S291A/T292A-mH (T292A) were collected, two hours after UV-irradiation (1 J/m$^2$, 10 J/m$^2$). To inhibit the cellular PLK1, BI2536 (2 µM) was added for 15 min, prior to the harvest. Cell lysates were subjected to immunoprecipitation using α-myc antibody-coated agarose beads. Western blotting analyses of the input (Input) and immunoprecipitates (IP) were performed, using α-RAD9 (RAD9), α-pT292 (pT292), α-pT313 (pT313), and α-pS326 (pS326) antibodies. See also *Figure 3—figure supplement 1*. PLK1 phosphorylates RAD9 on Thr 313 and Ser 326, *Figure 3—figure supplement 2*. PLK1 phosphorylates RAD9 on Thr 313 and Ser 326 in vivo and *Figure 3—figure supplement 3*. The phosphorylation status of RAD9 under various genotoxic stresses.

DOI: https://doi.org/10.7554/eLife.29953.006

The following figure supplements are available for figure 3:

**Figure supplement 1.** PLK1 phosphorylates RAD9 on Thr 313 and Ser 326.

DOI: https://doi.org/10.7554/eLife.29953.007

**Figure supplement 2.** PLK1 phosphorylates RAD9 on Thr 313 and Ser 326 in vivo.

DOI: https://doi.org/10.7554/eLife.29953.008

**Figure supplement 3.** The phosphorylation status of RAD9 under various genotoxic stresses.

DOI: https://doi.org/10.7554/eLife.29953.009

(pT313) and α-pSer326 (pS326), detected GST-RAD9 that had been incubated with PLK1 in kinase buffer (*Figure 3—figure supplement 1D*), and the western blotting signals were largely diminished when the mutant GST-RAD9 proteins, harboring either the T313A or S326A mutation, were used as substrates in the PLK1 kinase assay (*Figure 3—figure supplement 1E*). To confirm whether the phosphorylation at Thr313 and Ser326 occurred by PLK1 binding to RAD9, through the CDK-dependent phosphorylation of Thr292, the PLK1 kinase assay with GST-RAD9 was performed in the absence or presence of CDK2-cyclinA2. As demonstrated by western blotting, the appearance of both the pThr313 and pSer326 signals depended on the presence of CDK2-cyclinA2 (*Figure 3C*, quantification in *Figure 3—figure supplement 1F*). Furthermore, large decreases in the pThr313 and pSer326 western blot signals were observed when GST-RAD9-T292A was used as the substrate in the PLK1 kinase assay (*Figure 3D*, quantification in *Figure 3—figure supplement 1F*).

## PLK1 phosphorylates Thr313 and Ser326 of RAD9 in vivo

Next, we confirmed the PLK1-dependent phosphorylation of Thr313 and Ser326 of RAD9 in cells. First, to emphasize the PLK1 action on RAD9, PLK1 was ectopically expressed in asynchronously growing HEK293A cells that harbored a *CMV-tetO-RAD9-mH* construct (encoding RAD9-mH controlled by a doxycycline-inducible promoter) at the *FRT* locus. In addition, the phosphatase inhibitor calyculin A was added to the cells, prior to harvesting. The treatment with calyculin A forces the cells to enter into and arrest in M phase, a condition that should also activate PLK1 (*Coco-Martin and Begg, 1997*; *Ishihara et al., 1989*). The cell lysates were subjected to IP using α-myc antibody-coated agarose beads, and the immunoprecipitates were analyzed by western blotting. When the western blot was probed using the antibody against pThr313, the signals were seen with the expression of PLK1 (*Figure 3—figure supplement 2A*, lane 2). These signals completely disappeared when the PLK1-KR mutant was expressed and the cells were treated with the PLK1 inhibitor BI2536 (*Figure 3—figure supplement 2A*, lane 3). Together, these results suggested that PLK1 mediates the phosphorylation of Thr313 of RAD9. When the blots were instead probed with the antibody against pSer326, again the signals were detected when PLK1 was expressed, but these signals were decreased only slightly when PLK1-KR was expressed and the cells were treated with BI2536 (*Figure 3—figure supplement 2A*, compare lanes 2 and 3). This suggested that Ser326 can be phosphorylated by kinases other than PLK1. In this experiment, we also observed that the levels of both pThr313 and pSer326 were only slightly, if at all, decreased in the cells that expressed the RAD9-

S291A/T292A mutant (*Figure 3—figure supplement 2A*, lane 4), in contrast to the results obtained in the in vitro experiment. This distinction may be due to the strong activation and over-expression of PLK1 in calyculin A-treated cells, which may obviate the need for Thr292 phosphorylation to locally activate PLK1 for RAD9 phosphorylation.

Next, to determine whether endogenous PLK1 could phosphorylate RAD9, we employed asynchronously growing HEK293 cells harboring the *CMV-tetO-RAD9-mH* construct at the *FRT* locus. These cells were exposed to UV irradiation, in the presence or absence of the PLK1 inhibitor BI2536, before harvesting, and cell lysates were prepared. RAD9-mH was immunoprecipitated with α-myc antibody-coated agarose beads, and monitored by western blotting using α-RAD9, α-pThr292, α-pThr313, and α-pSer326 antibodies (*Figure 3E*). The signal recognized by α-RAD9 remained unchanged regardless of the UV irradiation dose, BI2536 treatment, or T292A mutation. However, the pThr292 signal, which was relatively unchanged in the UV-irradiated cells, was increased in the BI2536-treated cells (*Figure 3E*, IP; lanes 5 and 6), implying that this CDK-dependent phosphorylation event (pThr292) occurred upstream of the PLK1-dependent phosphorylation. We observed diminished pThr313 levels when the cells were treated with BI2536 (*Figure 3E*, lanes 5 and 6), or when T292A-mutated RAD9 was expressed (*Figure 3E*, lanes 7, 8 and 9) instead of wild-type RAD9 (WT). These results indicated that the generation of pThr313 is dependent on PLK1, which (as demonstrated above) is bound to and activated by pThr292 of RAD9. Interestingly, the pThr313 signal increased slightly when the cells were UV-irradiated (*Figure 3E*, lanes 3 and 4). We also observed that the pSer326 signal increased upon UV irradiation, and a slight decrease was detected when the cells were treated with BI2536 (*Figure 3E*, lanes 5 and 6). However, we did not see a decrease of pSer326 in the T292A-mutated RAD9 (*Figure 3E*, lanes 7, 8 and 9). Although (as shown above) pSer326 can be phosphorylated by PLK1, we postulate that other kinases can substitute for PLK1 for this function, permitting the phosphorylation of Ser326 without the prerequisite phosphorylation of Thr292.

Next, we analyzed whether DNA replicative stress could also affect the PLK1-dependent phosphorylation of RAD9. HEK293A cells harboring the *CMV-tetO-RAD9-mH* construct at the *FRT* locus were synchronized by a double-thymidine block, and the released cells were treated with 0.2 mM HU. The cell lysates were subjected to IP to monitor the phosphorylation of RAD9 (*Figure 3—figure supplement 2B*). Although the total amounts of (RAD9), pThr313 (pT313), and pSer 326 (pS326) were relatively unchanged by the HU exposure, the amount of pThr292 (pT292) was somewhat decreased. These data suggested that pThr292 efficiently triggers PLK1-dependent phosphorylation upon UV- or HU-induced stress.

We tested the effects of other types of DNA damage on the PLK1-dependent phosphorylation of RAD9. Treatments with camptothecin (CPT: topoI inhibitor), ionizing radiation (IR), or high doses of methylmethanesulfonate (MMS: 0.03%) or mitomycin C (MMC: 1 µg/ml) yielded slight increases in the pSer326 phosphorylation (*Figure 3—figure supplement 3A and B*). Aphidicolin (Aph: a DNA polymerase inhibitor) treatment reduced the levels of both pThr313 and pSer 326; however, pThr292 was also decreased, and thus these results suggested that the efficiency of PLK1-dependent phosphorylation is either maintained or slightly enhanced in response to most types of genotoxic stress (*Figure 3—figure supplement 3*).

## PLK1-dependent phosphorylation of RAD9 Thr313 is required to suppress S-phase checkpoint response

If PLK1 acts on RAD9 downstream of CDK in vivo, then the mutation of PLK1-dependent phosphorylation sites should affect cells in a manner similar to that seen with RAD9-T292A. We therefore constructed U2OS or HEK293A cell lines that encoded *RAD9* with the T313A or S326A mutation, and expressed the mutant proteins under the control of the *CMV* promoter from a construct located at the *FRT* locus. The expression of RAD9-mH in U2OS cells is shown in *Figure 4—figure supplement 1*.

First, we assessed whether the PLK1-dependent phosphorylation affected the retention of the mutated RAD9 proteins on damaged chromatin. In a fractionation assay, we observed the increased accumulation of T313A-mutated RAD9 in the insoluble fraction upon HU treatment (*Figure 4A*, Triton-X insoluble, +HU). We also observed the accumulation of Ser345-phosphorylated CHK1 in the insoluble fraction of HU-treated cells that expressed the T313A-mutated RAD9 (*Figure 4A*, Triton-X insoluble, +HU). It should be noted that lower levels of the S326A-mutated RAD9 accumulated in

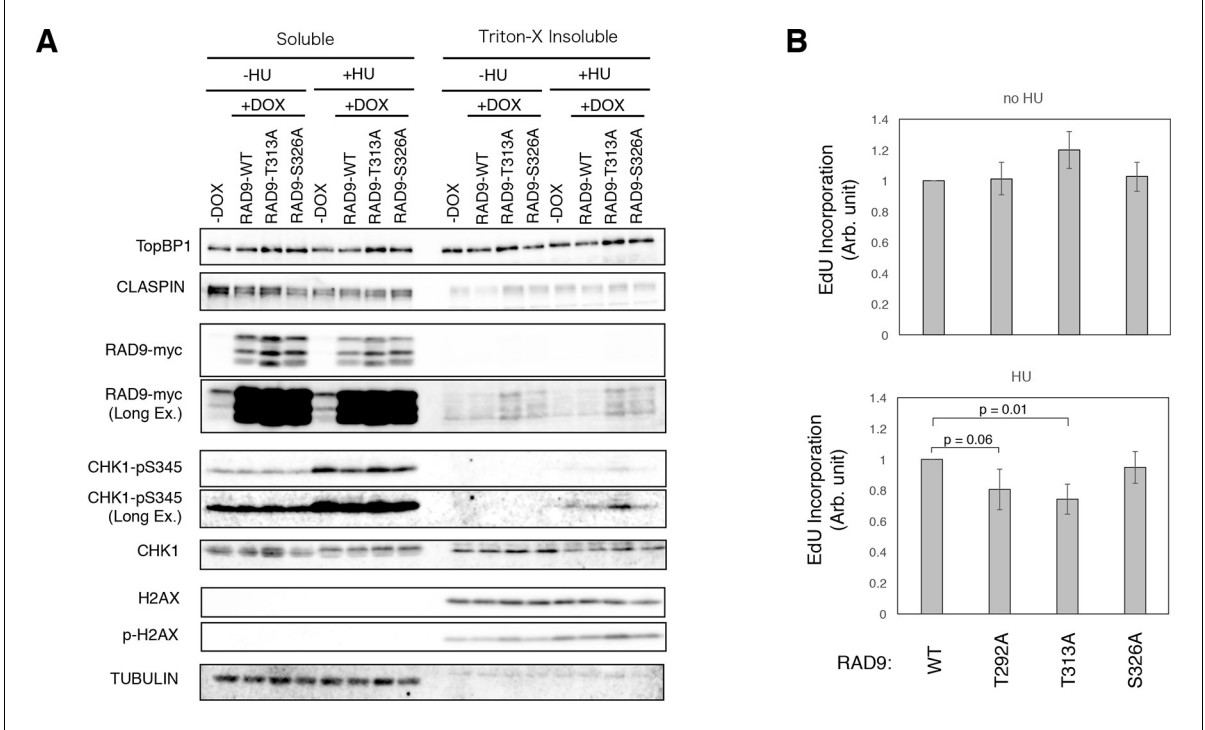

**Figure 4.** Enhanced checkpoint signaling when PLK1 fails to phosphorylate RAD9. (**A**) A chromatin fractionation assay (see Materials and methods) was performed in HEK293A-T-REx cell lines stably expressing RAD9-WT-mH (RAD9-WT), RAD9- T313A-mH (RAD9-T313A) or RAD9- S326A-mH (RAD9-S326A). Cells were grown in media containing 0.2 mM hydroxyurea for 24 hr (+HU). A western blotting analysis is shown. α-TopBP1, α-CLASPIN, α-myc (RAD9-myc), α-phospho-serine 345 of CHK1 (CHK1-pS345), α-CHK1, α-phospho-serine 139 of Histone H2AX (p-H2AX), α-H2AX, and α-tubulin were used. (**B**) The incorporation of ethynyl-deoxy-uridine (EdU) was quantified in U2OS cells stably expressing RAD9-WT-mH (WT), RAD9-S291A/T292A-mH (T292A), RAD9-T313A-mH (T313A) or RAD9-S326A-mH (S326A) from the *FRT* locus (see Materials and methods). The cells were incubated with 5 μM of EdU before the harvest (30 min for cells without hydroxyurea treatment, 1 hr for cells with 0.2 mM hydroxyurea). The samples were stained with Alexa Fluor 488 azide using the Click reaction, and subjected to the flow cytometry analysis. An example of the data is shown in *Figure 4—figure supplement 3*. The incorporated signals were quantified, and the mean values from three independent experiments are plotted. See also *Figure 4—figure supplement 1*. U2OS stable cell line that expresses RAD9-mycHis, *Figure 4—figure supplement 2*. DNA damage signaling was increased when PLK1 failed to phosphorylate RAD9, *Figure 4—figure supplement 3*. dNTP incorporation was decreased in HU treated cells where PLK1-dependent phosphorylation on RAD9 was defective, *Figure 4—figure supplement 4*. The analysis using the CMV-RAD9-expressing U2OS cell line under endogenous RAD9 knock down, *Figure 4—figure supplement 5*. Decreased dNTP incorporation was observed in telomerase-positive HEK293A cells that expressed CDK-PLK1-dependent phosphorylation defective RAD9 and *Figure 4—figure supplement 6*. EdU incorporation assay under Aphidicolin- and MMC-induced stress.

DOI: https://doi.org/10.7554/eLife.29953.010

The following figure supplements are available for figure 4:

**Figure supplement 1.** U2OS stable cell line that expresses RAD9-mycHis.
DOI: https://doi.org/10.7554/eLife.29953.011

**Figure supplement 2.** DNA damage signaling was increased when PLK1 failed to phosphorylate RAD9.
DOI: https://doi.org/10.7554/eLife.29953.012

**Figure supplement 3.** dNTP incorporation was decreased in HU treated cells in which PLK1-dependent phosphorylation of RAD9 was defective.
DOI: https://doi.org/10.7554/eLife.29953.013

**Figure supplement 4.** The analysis using CMV-RAD9-expressing U2OS cell line under endogenous RAD9 knock down.
DOI: https://doi.org/10.7554/eLife.29953.014

**Figure supplement 5.** Decreased dNTP incorporation was observed in telomerase-positive HEK293A cells that expressed CDK-PLK1-dependent phosphorylation defective RAD9.
DOI: https://doi.org/10.7554/eLife.29953.015

**Figure supplement 6.** EdU incorporation assay under Aphidicolin- and MMC-induced stress.
DOI: https://doi.org/10.7554/eLife.29953.016

the insoluble fraction (compared to the accumulation of RAD9-T313A under the same conditions). These results indicated that the PLK1-dependent phosphorylation of the Thr313 residue of RAD9 phenocopies the T292A-mutated RAD9, and suggested that PLK1 acts downstream of CDK to control the dissociation of RAD9 from chromatin upon genotoxic stress. Consistent with this interpretation, we also observed that the inhibition of the PLK1 activity by BI2536 also caused the accumulation of endogenous RAD9 in the insoluble fraction, when the cells were treated with HU (*Figure 4—figure supplement 2*). Since both TopBP1 and CLASPIN, which act as mediators for checkpoint signaling, also accumulated in the insoluble fraction upon the inhibition of PLK1 in HU-treated cells, we concluded that a PLK1-dependent phosphorylation event controls the association of the checkpoint machinery with chromatin.

Next, we analyzed whether the PLK1-dependent phosphorylation (pThr313) of RAD9 follows the CDK-dependent phosphorylation (pThr292), in terms of controlling the rate of DNA replication. To do so, we quantitated the rate of dNTP incorporation in the U2OS cells. The cells that expressed the T313A- or S326A-mutated RAD9 were exposed to 0.2 mM HU for 24 hr, and then to EdU for 1 hr prior to harvest. The incorporated EdU was fluorescently labeled by the click reaction, and the cells were subjected to a flow cytometric analysis. The cells that expressed the T313A-mutated RAD9 showed a significantly lower level of EdU incorporation, as compared to the cells that expressed wild-type RAD9 (as seen in T292A-RAD9 expressing cells) (*Figure 4B*, *Figure 4—figure supplement 3*). The cells that expressed the S326A-mutated RAD9 and those expressing wild-type RAD9 incorporated similar levels of EdU. These results suggested that pThr313, as well pThr292, of RAD9 is responsible for maintaining the rate of DNA replication upon genotoxic stress. Together, these experiments indicated that the CDK-PLK1 axis targets RAD9 to antagonize the checkpoint response upon replicative stress.

We also knocked down the endogenous RAD9 (*Figure 4—figure supplement 4A*), to determine whether the cellular effect observed upon the expression of the T313A- or S326A-mutated RAD9 is enhanced. In cells that expressed the T313A-mutated RAD9, the transfection of siRAD9 yielded an enhancement in the phosphorylation of Ser345 of CHK1 (*Figure 4—figure supplement 4B*) and a slight reduction in the rate of DNA replication upon HU treatment, as compared to cells that were not treated with siRAD9 (*Figure 4—figure supplement 4C*). These results indicated that, although the expression of RAD9 under the control of the *CMV* promoter was sufficient to confer the phenotype, the knockdown of the endogenous RAD9 could enhance the cellular effect to a certain extent. In cells that expressed the S326A-mutated RAD9, the silencing did not result in an obvious enhancement of the phosphorylation of Ser345 of CHK1 or a decrease in the DNA replication rate (*Figure 4—figure supplement 4D,E*). Furthermore, we also confirmed the reduction in the DNA replication rate in telomerase-positive HEK293A cells (*Bryan et al., 1995*) expressing either T292A- or T313A-mutated RAD9 (*Figure 4—figure supplement 5A and B*). This result excluded the possibility that the slowing of the DNA replication rate was caused by an atypical mechanism for checkpoint control, which may occur in U2OS cells via Alternative Lengthening of Telomeres (ALT) (*Figure 4—figure supplement 5A and B*).

A slower DNA replication rate was also observed in response to other types of DNA damage. Specifically, aphidicolin treatment also reduced the rate of EdU incorporation, and MMC treatment caused more cells to accumulate in early S phase when the T313A-mutated RAD9 was expressed, although the latter treatment did not yield an obvious decrease in the EdU incorporation (*Figure 4—figure supplement 6*). The observed phenotype may have been specific to the direct inhibition of DNA replication by some agents (HU, a dNTP synthesis inhibitor; aphidicolin, a DNA polymerase inhibitor), but not by others (MMC, a DNA damaging agent that crosslinks DNA strands).

We next performed a DNA combing assay, to determine which steps of DNA replication contribute to slowing down bulk DNA synthesis. The DNA damage checkpoint involves controlling either the dormant/late replication origin firing or the velocity of individual replication forks. The DNA combing technique allows the measurement of individual replication fork rates, and of the IOD (*I*nter-*O*rigin *D*istance) values that may reflect the efficiency of dormant origin firings. At the end of the HU treatment, the WT- or T313A-mutated-RAD9 expressing U2OS cells were pulse-labeled by 5-iodo-2'-deoxyuridine (IdU), preceded by 5-chloro-2'-deoxyuridine (CldU) (*Figure 5A*). The DNA synthesis on DNA fibers was visualized (*Figure 5B*), and the IOD values and the fork velocities were measured (*Figure 5C,D,E*). In the absence of HU, the rates of individual replication forks in cells that expressed RAD9-WT were 0.85 ± 0.36 kb/minutes, and those of the RAD9-T313A expressing cells

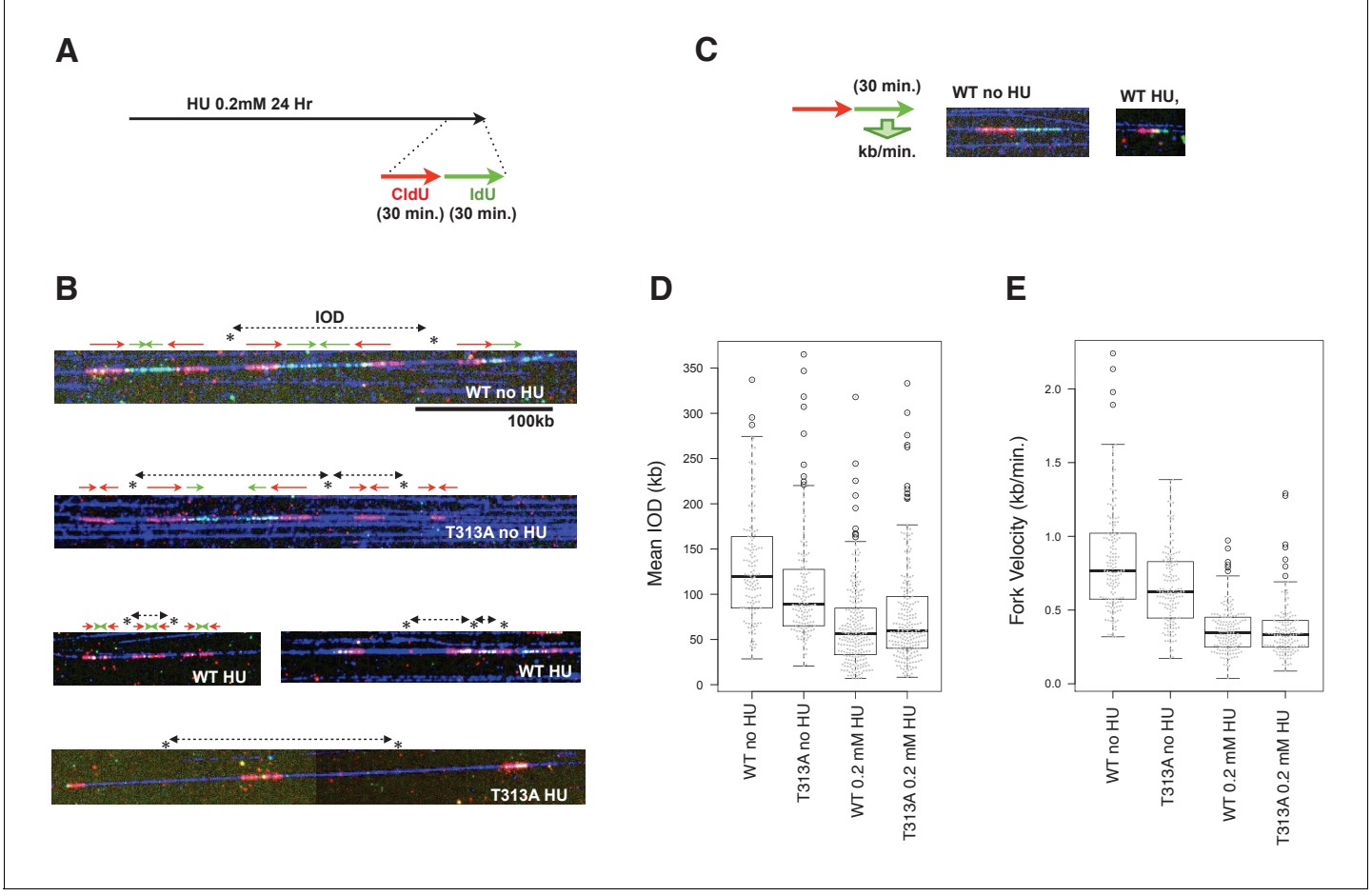

**Figure 5.** The effect of origin firing and DNA replication fork integrity of PLK1-dependent phosphorylation on RAD9. (**A**) A schematic of the time course applied for the combing assay analysis. U2OS cells expressing WT- or T313A-RAD9 from genomic *FRT* sites were used for the analysis. (**B**) Examples of DNA fibers seen in the DNA combing assay. The vertical strips seen were either the boundary of the assembled photo frames or noise produced upon scanning of DNA fibers. See the supplement for the surrounding regions (***Supplementary file 3***). (**C**) A schematic of the measurement of replication fork velocity. A green signal (IdU labeled DNA) was measured on a unidirectionally moving fork that is associated with a red signal (CldU labeled DNA). (**D**), (**E**) Mean *Inter Origin Distance* (IOD) values were measured and plotted on the graph in (**D**) (*WT no HU*: n = 140, *T313A no HU*: n = 146, *WT 0.2 mM HU*: n = 231, *T313A 0.2 mM HU*: n = 229). The velocity of each DNA replication fork was measured and plotted in (**E**). Boxplots overlaid with beeswarm plots are shown. Boxes indicate the upper and lower quartiles, and the central bar indicates the median value of the measurements. The graphs were made with the R program. The *p*-values in the IOD analysis were $8 \times 10^{-8}$ (without HU) and 0.058 (with HU) (WT vs. T313A; the measurement values that were statistically significant (within the top and bottom whisker bars) were applied to calculate the *p*-values). The *p*-value in the replication fork velocity analysis was $5.8 \times 10^{-7}$ (WT vs. T313A, without HU). See also ***Figure 5—figure supplement 1***. The DNA combing assay in aphidicolin-treated cells, ***Figure 5—figure supplement 2***. Enhanced RAD9-CLASPIN complex formation when CDK or PLK1 failed to phosphorylate RAD9.

DOI: https://doi.org/10.7554/eLife.29953.017

The following figure supplements are available for figure 5:

**Figure supplement 1.** The DNA combing assay in aphidicolin-treated cells.
DOI: https://doi.org/10.7554/eLife.29953.018
**Figure supplement 2.** Enhanced RAD9-CLASPIN complex formation when CDK or PLK1 failed to phosphorylate RAD9.
DOI: https://doi.org/10.7554/eLife.29953.019

were slower (0.66 ± 0.27 kb/minutes, *p*-value: $5.8 \times 10^{-7}$). The IOD values were 131 ± 62 kb in RAD9-WT expressing cells, and 107 ± 61 kb in RAD9-T313A expressing cells. This shorter IOD distance in Rad9-T313A expressing cells implied a higher efficiency of origin firing, which could result in a similar rate of bulk DNA synthesis to that seen in RAD9-WT expressing cells under non-stressed conditions. We then measured the fork rates under the HU stress, and observed that the rates of the DNA replication forks became slower in both RAD9-WT and RAD9-T313A expressing cells, with

similar velocities (WT: 0.37 ± 0.16 kb/minutes, T313A: 0.37 ± 0.19 kb/minutes). Under these conditions, the IOD of the RAD9-T313A expressing cells was slightly longer than that of RAD9-WT expressing cells (WT: 61 ± 38 kb, T313A: 69 ± 41 kb), although the difference is marginally significant (*p*-value: 0.058).

The replication fork velocities under non-stressed and stressed conditions were confirmed in an independent combing assay. Samples for either non-treated or aphidicolin-treated (0.4 µM) cells (RAD9-WT and -T313A expressing cells) were applied. A slower replication fork progression rate was observed in the non-treated RAD9-T313A expressing cells (*Figure 5—figure supplement 1*), consistent with the data obtained in the former assay (*Figure 5*), although the absolute values of the fork progression rate were smaller under the latter conditions, perhaps due to the different method used to prepare the fibers on the slides (see Materials and methods). When the cells were treated with aphidicolin, the replication fork rate was similar to that under the conditions with 0.2 mM HU, again suggesting that the fork progression rate did not cause the difference in the slow S-phase progression rate. In summary, we concluded that the reason for the slower rate of S-phase progression is not due to the reduction of the replication fork rate in the RAD9-T313A expressing cells. Further analysis is needed to determine whether the increase in the distance between firing origins contributes to the slow S-phase. It should also be noted that the replication fork rate was already slower in the RAD9-T313A expressing cells under non-stressed conditions, which may suggest that the checkpoint response is enhanced in the RAD9-T313A expressing cells in the normal cell cycle (see *discussion*).

Finally, we observed that the T292A- or T313A-mutated RAD9 co-immunoprecipitated with increased amounts of CLASPIN upon HU treatment ((*Figure 5—figure supplement 2*), see 'HU' in IP). Together, these results suggested that the S-phase checkpoint response is enhanced when PLK1 fails to phosphorylate RAD9.

## CDK- and PLK1-dependent phosphorylation of RAD9 allows cells to grow under genotoxic stress

If the collaboration of CDK and PLK1 in phosphorylating RAD9 is required for decreasing RAD9's S-phase checkpoint function, then cells expressing the T292A- or T313A-mutated RAD9 should be growth-compromised under genotoxic stress. Indeed, we observed that cells expressing the T292A- or T313A-mutated RAD9 exhibited a slower growth rate in the presence of HU (For U2OS cells, *Figure 6A*; for HEK293A cells, *Figure 6—figure supplement 1*). Additionally, as compared to the cells that expressed wild-type RAD9, the cells that expressed T313A-mutated RAD9 exhibited a significant decrease in colony-forming efficiency (as assessed by a cell viability assay) upon exposure to 0.2 mM HU (*Figure 6B*). Together, these results imply a role for the CDK- and PLK1-dependent phosphorylation of RAD9 in facilitating growth under genotoxic stress.

## Discussion

Cells must overcome environmental stresses that have the potential to disturb their proliferation. In the case of genotoxic stress, DNA damage checkpoints are activated to restore cellular viability. However, such checkpoints can also serve as anti-proliferation signals. Thus, the proliferation status of cells must be coupled to the activities of the checkpoint machineries. Here, we demonstrated the collaborative action of CDK and PLK1 in phosphorylating the checkpoint protein RAD9, thereby minimizing RAD9's checkpoint activating function to maintain the rate of S-phase, and in turn to maintain the higher cellular growth rate (*Figure 7*). The CDK- and PLK1-dependent phosphorylation of RAD9 reduces the binding of RAD9 to damaged chromatin, suggesting the dynamic control of active checkpoint complexes at DNA damage sites to regulate the degree of DNA damage checkpoint signaling. Our results have elucidated the mechanism by which the cell cycle status controls the DNA damage sensor complex to ensure proliferation under genotoxic stress, thus improving our understanding of the balance between anti- and pro-proliferation signals.

## CDK and PLK1 target RAD9 to modulate DNA damage detection

When cells that were defective in the CDK- or PLK1-dependent phosphorylation of RAD9 (demonstrated by the expression of T292A- or T313A-mutated RAD9) were grown under replicative stress (0.2 mM HU), more cells accumulated in S phase, and the incorporation of dNTPs during S phase

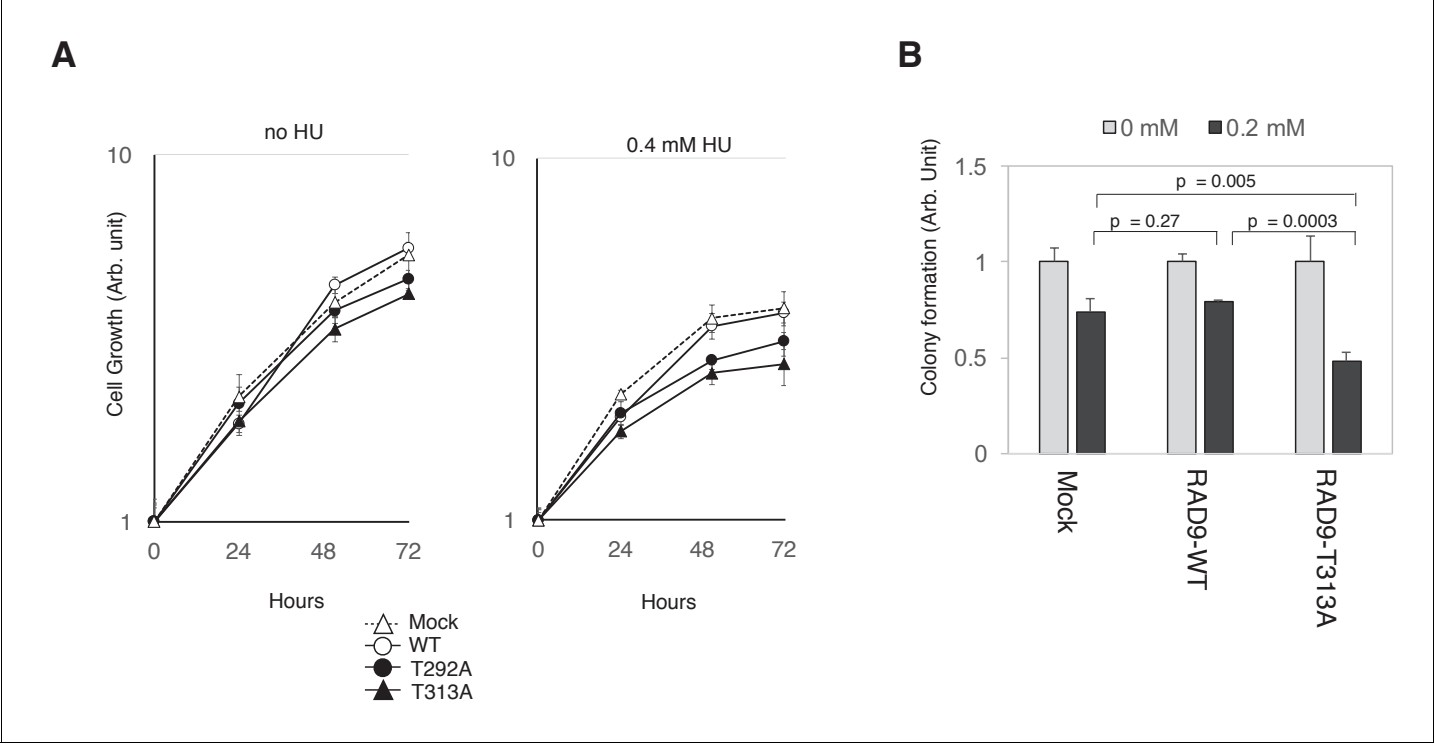

**Figure 6.** PLK1-dependent phosphorylation of RAD9 is required for proliferation under replicative stress. (**A**) Cell growth was monitored using the Presto-Blue Cell viability reagent (see Materials and methods). U2OS cells stably expressing RAD9-WT-mH (WT), RAD9-S291A/T292A-mH (T292A), and RAD9-T313A-mH (T313A) from the *FRT* locus, and the host U2OS T-REx cells (Mock) were used. (**B**) The cellular viability in the presence of hydroxyurea was monitored by a colony formation assay (see Materials and methods). U2OS cells stably expressing RAD9-WT-mH (RAD9-WT) and RAD9-T313A-mH (RAD9-T313A) from the *FRT* locus, and the host U2OS T-REx cells (Mock) were used. See also *Figure 6—figure supplement 1*. Reduced cell proliferation by expression of non-phosphorylatable RAD9 in telomerase-positive HEK293A cells.
DOI: https://doi.org/10.7554/eLife.29953.020

The following figure supplement is available for figure 6:

**Figure supplement 1.** Reduced cell proliferation by expression of non-phosphorylatable RAD9 in telomerase-positive HEK293A cells.
DOI: https://doi.org/10.7554/eLife.29953.021

was greatly decreased, as compared to that observed in cells expressing wild-type RAD9. These results suggested that CDK plays a role in maintaining a high rate of S-phase progression, via the facilitation of the PLK1-dependent phosphorylation of RAD9.

The region where CDK phosphorylates RAD9 (on Thr292) has a typical consensus sequence for a PBD binding site (*van Vugt et al., 2010*). Using a peptide competition assay, we showed that PLK1 bound RAD9 phosphorylated at Thr292. Furthermore a peptide phosphorylated on Thr292 activated PLK1. Since PBD acts as an auto-inhibitory domain, with inhibitory activity relieved when the domain is bound to a substrate (*Xu et al., 2013*), we concluded that RAD9 phosphorylated on Thr292 provides a genuine binding site for PLK1.

The interaction between a PBD and the phospho-peptide is reportedly strong, with a dissociation constant in the range of a few hundred nanomolar (*van Vugt et al., 2010*). This high affinity may allow a minimal amount of PLK1 to target the Thr292-phosphorylated RAD9, and would suit binding by PLK1, which exhibits rather repressed expression during interphase (*Winkles and Alberts, 2005*). In this scenario, once Thr292 is phosphorylated, PLK1 easily targets and binds RAD9, leading to the local activation of PLK1 and the phosphorylation of the Thr313 residue of RAD9.

What is the molecular impact of the PLK1-dependent phosphorylation of RAD9? RAD9, as a component of the PCNA-like 9-1-1 complex, is loaded onto 5' recessed DNA ends at damage sites, and also accommodates TopBP1 and CLASPIN to facilitate signaling through checkpoint kinases (*Delacroix et al., 2007*; *Furuya et al., 2004*; *Kumagai et al., 2006*; *Lee et al., 2007*; *Liu et al., 2012*; *Majka et al., 2006*; *Navadgi-Patil and Burgers, 2009*; *Takeishi et al., 2010*). We speculate

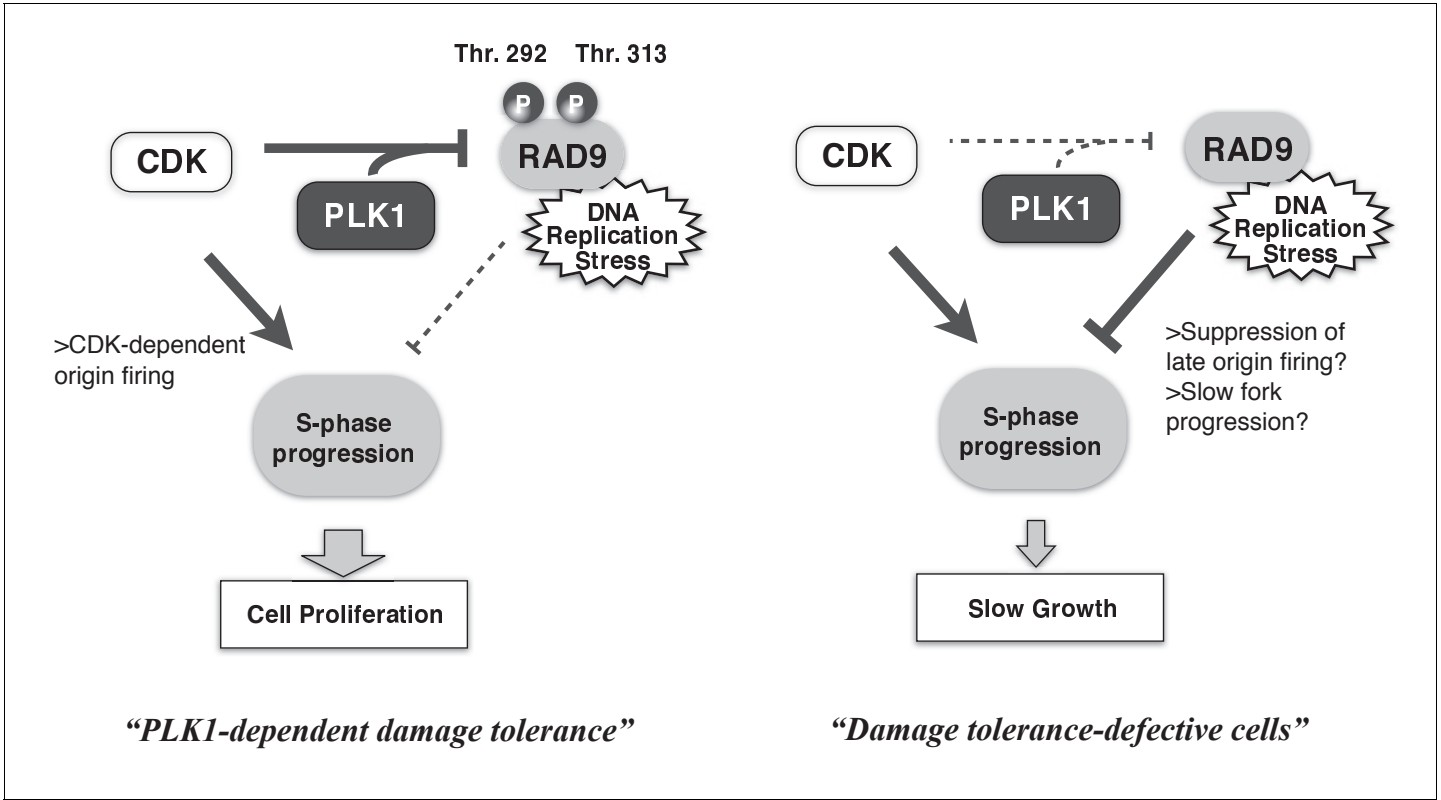

**Figure 7.** A schematic of PLK1-dependent damage tolerance. (*Left*) The collaboration of CDK and PLK1 targets and suppresses the DNA damage detection machinery in the DNA damage checkpoint process, and promotes the S-phase progression to support cell proliferation. (*Right*) When CDK and PLK1 fail to phosphorylate RAD9 (in the case of the PLK1 inhibitor, RAD9-T292A or -T313A mutant expressing cells), DNA damage checkpoint activation is increased and confers the slow growth.

DOI: https://doi.org/10.7554/eLife.29953.022

that RAD9's function as a DNA damage sensor is the target of PLK1-dependent regulation, given that we observed increased amounts of RAD9 on the chromatin when PLK1 failed to phosphorylate RAD9. The PLK1-dependent phosphorylation of RAD9 would minimize the amount of RAD9 bound at sites of stalled replication forks or replicative DNA damage, possibly by giving dynamism to the 9-1-1 complex from the DNA ends, and may in turn reduce the number of active checkpoint complexes. In RAD9, the sites of PLK1-dependent phosphorylation are located within the tail region. This region of RAD9 is expected to help the protein associate with DNA damage sites. For example, the region phosphorylated by PLK1 is close to the region required for the association with the single-stranded-DNA binding protein RP-A (*Figure 3—figure supplement 1A*)(*Xu et al., 2008*). Furthermore, we did not observe a decrease in the amount of the RAD9-bound clamp-loader RAD17 in the co-IP experiment when T292 was mutated, even under genotoxic stress conditions (data not shown). Thus, the PLK1-dependent phosphorylation may affect the affinity of the 9-1-1 complex itself to the DNA damage sites. This regulation is influenced by the balance between the cellular pro-mitotic activity and the DNA damage checkpoint activity, which may be determined by the amounts of PLK1, the CDK activity, and the degree of DNA damage stress. In other words, the cellular proliferation status may influence the cell's ability to detect DNA damage during checkpoint activation.

## Phosphorylation of the checkpoint sensor protein by CDK and PLK1 is required for maintaining the rate of DNA replication upon genotoxic stress

In eukaryotic cells, the rate of S-phase progression is related to the number of available replication origins (*Yekezare et al., 2013*). Upon growth under genotoxic stress, the replication rate can be controlled by S-phase checkpoint signaling, and when the signaling is dominant, it inhibits the firing

of the late-replication origins (*Santocanale and Diffley, 1998*; *Shirahige et al., 1998*). This inhibition of origin firing might prevent the generation of additional replicative stress in un-replicated regions of the genome, and would suppress genomic instability (*Boos et al., 2011*; *Petermann et al., 2010*; *Yekezare et al., 2013*). Alternatively, the slow S phase progression rate could be caused by the reduction in the progression rate of individual replication forks, which is reportedly dependent on CHK1 activation upon camptothecin-induced stress, and may allow the fork to mount a more robust response to the damage (*Iyer and Rhind, 2013*; *Seiler et al., 2007*). It has also been proposed that the reduction of the fork speed results in the activation of dormant origins proximal to the stalled replication fork, a process that may compensate for decreases in the local rate of DNA replication (*Yekezare et al., 2013*).

The DNA fiber/combing assay under the HU treated conditions revealed that the velocities of the replication forks were the same between the WT- and T313A-RAD9 expressing cells. This could indicate that the slower S-phase progression seen in the T313A-mutated cells was not caused by the slower velocity of the DNA replication forks. We observed a slight increase in the IOD value in the T313A-RAD9 expressing cells, which implies the inefficient firing of dormant origins in these cells. However, its marginal significance (p-value=0.058; WT vs. T313A) may indicate another reason, such as the contribution of the inhibition of late origin firing to the slower progression of S-phase in T313A-expressing cells under HU-induced stress. Perhaps more significantly, even under non-stressed conditions, we observed a slower DNA replication fork velocity in the T313A-RAD9 expressing cells, as compared to that in WT-RAD9 expressing cells. This could indicate that a moderate activation of the S-phase checkpoint exists even in the normal cell cycle process in the T313A-RAD9 expressing U2OS cells. In this case, the efficient firing of dormant origins may compensate for the overall S-phase rate; however, a more detailed analysis (e.g., shorter pulse-label duration) should be performed to arrive at a definite conclusion. In the T313A-expressing cells, further exposure to HU-induced stress would activate an excessive DNA damage response and could cause a reduction in the number of firing origins, which would result in a slower rate of S-phase progression. In this scenario, PLK1 may raise the threshold of the DNA damage level, which would require more DNA damage stress for the checkpoint activation during S-phase. In other words, the PLK1-dependent phosphorylation of RAD9 may suppress the activation step of the checkpoint, rather than diminishing the activated DNA damage checkpoint processes, and this process could act as a novel DNA damage tolerance mechanism to facilitate cell proliferation. The dynamics between the proliferation control and the DNA damage sensors at DNA damage sites should be clarified in the future, to facilitate our understanding of the ability of cells to drive proliferation under environmental stresses.

## PLK1 mediates tolerance to genotoxic stress

PLK1 is a pro-mitotic kinase, and its main function is to facilitate the mitotic process. However, PLK1 also functions in DNA metabolism, and probably coordinates cellular proliferation and genomic stability. In addition to its role in antagonizing the DNA damage checkpoint activity, PLK1 reportedly promotes DNA replication by phosphorylating Orc2, an event that may be critical to maintain genomic DNA replication under replicative stress (*Song et al., 2011*). The PLK1-mediated phosphorylation of Ser188 of Orc2 is responsible for the resistance of pancreatic tumor cells to the chemotherapy drug gemcitabine, an agent that inhibits DNA replication (*Song et al., 2013*). This phosphorylation event probably stabilizes the pre-replicative complexes to ensure that the cells continue to progress through S phase despite the genotoxic stress, possibly by keeping dormant origins active (*Song et al., 2013*). Combined with our finding of the PLK1-dependent phosphorylation of RAD9, these results suggest that PLK1 employs multiple targets to drive S-phase progression in cells under stress, thus facilitating tolerance to genotoxic stresses.

For cancer cells, counteracting these checkpoint-related processes is expected to be a favorable tactic for driving cellular proliferation. Indeed, the ectopic expression of PLK1 is associated with a wide range of tumor types. PLK1 is also known to be involved in promoting resistance to chemotherapeutic regimens with drugs such as doxorubicin (a DNA intercalating compound) or gemcitabine (a DNA analogue that serves as an inhibitor of DNA replication) (*Gutteridge et al., 2016*). Thus, an improved understanding of DNA-related metabolism involving PLK1 will be met with keen interest, as such information could facilitate the effective use of PLK1 inhibitors during cancer therapies.

In summary, we demonstrated that CDK and PLK1 target the DNA damage sensor RAD9 to minimize the checkpoint response, and revealed an underlying mechanism whereby cells can tolerate

genotoxic stress and proceed with proliferation. Thus, a complex balancing mechanism exists to coordinate proliferation and the stress response, and more systematic investigations of this mechanism should be conducted in the future.

## Materials and methods

### Plasmids and primers

The *RAD9A* cDNA was purchased from OriGene. The gene encoding C-terminally myc-His6 (mH)-tagged *RAD9* was constructed in pCDNA3.1 or pCDNA5/FRT. The cloned *RAD9* was mutagenized for siRAD9 (s11719: Ambion) resistance. The mutations in *RAD9* were introduced using KOD polymerase (TOYOBO, Japan) and mutagenized primers. The primers used to mutagenize *RAD9* are presented in the supplementary table (*Supplementary file 1*). The primers to construct the resistant *RAD9* against siRAD9 (s11719: Ambion) were #2147/#2150, and are also included in *Supplementary file 1*. The gene encoding GST-tagged *RAD9* used for bacterial expression was constructed on the pGEX-6P plasmid. Primers #1663/#1664 (*Supplementary file 1*) were used for the subcloning. For the two-hybrid assay, a pLex-A fusion system and pACT2 (or pGADT7) were used to construct the bait and prey plasmids, respectively (*Bartel and Fields, 1995*; *Tanaka et al., 2013*). The regions encoding the polo-box domains of *PLK1-5* were amplified from a cDNA library (a kind gift from Dr. Seiji Tanaka, Kochi University of Technology), using the primers listed in *Supplementary file 1*. Other plasmids used to express the PLK1, pCDNA-flag-PLK1 and pGEX-6P-PLK1 (wt and K82R) proteins were kind gifts from Dr. M. Osugi (Tokyo University). For the expression of the active CDK protein, the plasmid that expresses the CIV1 (CAK in *S. cerevisiae*), CDK2, and Cyclin E proteins, controlled by a dual promoter system, was provided by Dr. Tatsuro Takahashi (Kyushu University), and the plasmid that expresses the active CDK2-CyclinA2 protein is a derivative of that plasmid.

### Cell culture and transfection

HEK293A cells (purchased from Invitrogen) and HEK293A T-REx cells were cultured in Dulbecco's modified Eagle's medium, supplemented with 10% fetal bovine serum. U2OS cells (purchased from IDAC, Tohoku University) and U2OS T-REx cells were cultured in RPMI medium, supplemented with 10% fetal bovine serum. Cell lines used were exposed to plasmocin to avoid possible contamination of mycoplasma and tested for the TAKARA PCR mycoplasma kit. Plasmid transfection was performed using the Attracten transfection reagent, according to the manufacturer's protocol (Qiagen). For the induction from the tetO-harboring plasmid, doxycyclin (0.5 μg/ml) was used. For the knockdown of RAD9, siRAD9 (Ambion: s11719) was used. The transfection of the siRNA was performed using the Dharmafect one transfection reagent, according to the manufacturer's protocol (Dharmacon). The control siRNA was purchased from QIAGEN. To inhibit cellular kinases, a PLK1 inhibitor (BI2536: AdooQ Bioscience, USA) and a CDK2 inhibitor (RO-3306: Santa Cruz, USA) were used. To perturb cell cycle progression, hydroxyurea (Sigma-Aldrich) was used. For the double thymidine block, first, 2.5 mM thymidine was added to the media, followed by 10 hr of incubation without thymidine. Then, a second thymidine block was performed for 20 hr before the release.

### Cell growth or viability assay

To monitor cell growth, the Presto-Blue Cell Viability Reagent (Thermo Fisher) was used according to the manufacturer's protocol. Briefly, $1 \times 10^4$ cells / 150 μl / well of U2OS cells that stably express the *RAD9* gene were cultured in 96-well dishes with hydroxyurea. Every 24 hr, the 96 well dish was monitored for cell growth. Presto-Blue (17 μl) was added to each well, followed by an incubation in a $CO_2$ incubator at 37 °C for 10 min. Subsequently, 75 μl of 10% SDS was added to stop the reaction, and the plate was placed in an ARVO-X3 plate reader (Perkin Elmer). A 531 nm excitation filter and a 616 nm emission filter were used, and for each cell line and hydroxyurea condition, triplicate samples were measured and the average values were plotted in the graph. To monitor the cell viability upon hydroxyurea exposure, a colony formation assay was performed. First, U2OS cells that express the RAD9 protein were treated with 0.2 mM hydroxyurea for 36 hr. The cells were collected by trypsinization, and 500 cells / 10 ml were plated onto 10 cm dishes, followed by an incubation at 37 °C in a $CO_2$ incubator for 11 days. The plates were stained with crystal violet/methanol.

## Antibodies and western blotting

Western blot analyses were performed with anti-γ-H2AX (Millipore), anti-H2AX (Sigma-Aldrich), anti-RAD9 (Santa Cruz or Bethyl), anti-RAD17, anti-RAD1, anti-TopBP1, anti-CLASPIN (Santa Cruz), anti-tubulin, anti-CHK1, anti-CHK1-pS345, and anti-CHK2-pT68 (Cell Signaling Technology) antibodies. The antibodies against the phospho-peptides (pThr292: LQAHSpTPHPDC, pThr313: CAMETp-TIGNEG, pSer326: CEGSRVLPSIpS) were raised by MBL (Medical and Biological Laboratories (MBL), Japan). HRP-conjugated antibodies were used as secondary antibodies, the ECL signal was incorporated using a Licor Odyssey Fc imager system, and the quantification was performed with the Image Studio software.

## Flow cytometry and EdU incorporation assay

To measure the incorporation rate of deoxyribonucleotides, the amount of incorporated EdU in cells was measured by a flow cytometer using the reported labeling protocol (*Jia et al., 2015*). Cells were incubated with 5 μM of EdU (for the non-stressed cells: 30 min, for the HU-stressed cells: 60 min) before fixation by formalin (3.7%) for 10 min on ice. The PBS-washed cells were subjected to methanol fixation. The fixed cells were washed twice with PBS-Triton X-100 (0.1%), and subjected to the Click reaction for one hour at room temperature. The reaction mixture included 20 mM Tris (pH 7.4), 1/1,000 Alexa Fluor 488-azide (Molecular Probes), 0.1% $CuSO_4$, and 1 mg/ml of ascorbate. The labeled cells were washed with PBS-Triton X-100 and treated with a PI/RNaseA solution for 30 min, followed by a wash with PBS-Triton X-100. The cell cycle and the fluorescently labeled EdU were measured using a Guava easyCyte (Millipore) or FACSCalibur flow cytometer.

## In vitro assays

The proteins were purified through GSH-beads (4 °C, 1 hr in TEG buffer (Tris-Cl, pH 7.5, 50 mM, NaCl 150 mM, EDTA-Na 10 mM, glycerol 10%, Triton X-100 0.5–1%)). The GST-RAD9 proteins were eluted by 10 mM of glutathione (15 min, 3 times), and the CDK complex and PLK1 proteins were eluted by PreScission Protease (GE Healthcare, 15 min, 3 times). After the elution of the CDK complex and PLK1, the proteins were further incubated with GSH-beads to eliminate the residual GST-tagged proteins. All of the purified proteins were subjected to dialysis in 50% glycerol, 10 mM Tris-Cl (pH 8.0). The in vitro kinase assay using the recombinant active CDK complex or PLK1 was performed in kinase buffer (50 mM HEPES, pH 7.4, 10 mM $MgCl_2$, 1 mM DTT) supplemented with 1 mM ATP. The GSH pull down assay using GST-RAD9 was performed in kinase buffer supplemented with 0.1% Triton X-100.

## Immunoprecipitation

Cell lysates were prepared in HEPES-based buffer (20 mM HEPES, pH 7.4, 150 mM NaCl, 10% glycerol, 0.1 % NP40, 0.1% Triton X-100, 1 mM DTT, 0.5 mM PMSF). The lysates were sonicated and the soluble fractions were quantified using a protein assay kit (Bio-Rad), and adjusted to 2 μg/μl. The immunoprecipitation was performed using a c-Myc tagged Mild Purification Kit (MBL).

## Yeast two-hybrid assay

To quantify the beta-galactosidase activity, an ONPG (SIGMA-Aldrich) assay was performed, according to the Yeast Protocol Handbook (Clontech).

## Liquid chromatography-tandem mass spectrometry (LC-MS/MS)

Coomassie Brilliant Blue-stained protein bands were excised from the polyacrylamide gel and subjected to in-gel tryptic digestion. The digestion was performed using an In-Gel Tryptic Digestion Kit (Thermo Fisher Scientific, Waltham, MA), according to the manufacturer's instructions. The digestion products were reconstituted in 0.1% formic acid and injected for separation into a NanoLC-Ultra 2D-plus HPLC equipped with a cHiPLC Nanoflex system (Eksigent, Dublin, CA), which was used in the trap and elute mode with a trap column (200 μm x 0.5 mm, ChromXP, C18-CL, 3 μm, 120 Å, Eksigent) and an analytical column (ChromXP, C18-CL, 3 μm, 120 Å, Eksigent). The separation was performed with a binary gradient of solvent A (98% water, 2% acetonitrile, 0.1% formic acid) and solvent B (20% water, 80% acetonitrile, 0.1% formic acid). The gradient program was 2% to 40% solvent B for 125 min, 90% solvent B for 5 min, and 2% solvent B for 19 min, at a flow rate of 300 nL/

min. The eluates from the Nano-LC system were infused on-line to the TripleTOF 5600 + mass spectrometry system (SCIEX, Framingham, MA) equipped with a NanoSpray III ion source and a heated interface. The data sets were acquired with the information-dependent acquisition method.

## Identification of the phosphorylated peptides and quantitative estimation of the specific phosphorylation

The identification of the phosphorylated peptides was accomplished using the ProteinPilot software (version 4.5 Beta, SCIEX). The database was UniProtKB/Swiss-Prot (June 2014, Human), appended with a known contaminant database (SCIEX) and the GST-Rad9C fusion protein sequence. The special factors were gel-based ID and phosphorylation emphasis. Biological modifications were taken into account. The quantitative estimation of the specific phosphorylation was performed by the spectral counting method. The number of reliable MS/MS spectra (confidence $\geq$95%) corresponding to the peptide groups derived from the GST-Rad9 C-terminus, which had phosphorylation on the same site, irrespective of the other co-existing post-transcriptional modifications, were counted in each group and used for the quantitative estimation.

## DNA molecular combing assay

The DNA fiber assay was performed as described (*Conti et al., 2007*; *Kuriya et al., 2015*; *Michalet et al., 1997*). Briefly, cells were labeled with 100 µM of IdU or CldU for 30 min and harvested by trypsinization. The harvested cells were embedded in agarose plugs and treated overnight with proteinase K (2 mg/ml) in ESP buffer (1% N-L-sarcosyl in 0.5 M EDTA). The prepared samples were subjected to DNA fiber assays to monitor the DNA replication fork rate (stretching factor upon combing was 1.95 kb/µm). For the DNA combing assay to monitor the Inter-Origin-Distance, the samples were sent to GENOMIC VISION (France) for the analysis. The samples were prepared as advised, and two slides per sample were analyzed (stretching factor upon combing is approx. 2 kb/µm). The Inter-Origin-Distance was calculated using Fiber Studio (GENOMIC VISION). We noticed that the values of the replication fork tracks were longer in the samples obtained from GENOMIC VISION. These were partly caused by the difference in the buffer or glass that was used to stretch the DNA fibers. Thus, we emphasize the relative differences in values within the samples obtained from same series of experimental preparations.

## Acknowledgements

We thank all of members of the Radiation Biology Center for helpful comments and suggestions. Especially, we thank Ms. Mika Gunji for technical contributions. We thank Dr. Miho Ohsugi (University of Tokyo) for providing the PLK1 plasmids and the tips for PLK1 purification, Dr. Tatsuro Takahashi for providing the CDK plasmid, and Dr. Seiji Tanaka (Kohchi University of Technology) for providing the yeast two hybrid system and advice. This work was supported by KAKENHI (Grant-in-Aid for Scientific Research) for Young Scientists (A): (24687023 to KF), Grants-in-Aid for Scientific Research on Innovative Areas (23131515, 25131707 to KF), (C): 17K07284 (KF), The Uehara Memorial Foundation Grant (KF) and The Takeda Science Foundation grant for life science (KF). This work was also partly supported by NIG-JOINT (National Institute of Genetics, 2011–2017-A, (KF)), the Cooperative Research Project Program of Joint Usage/Research Center at the Institute of Development, Aging and Cancer, Tohoku University (2017, (KF)) and the joint travel program of the Institute of Molecular Embryology and Genetics, Kumamoto University (2017, (KF)).

## Additional information

### Funding

| Funder | Grant reference number | Author |
|---|---|---|
| Ministry of Education, Culture, Sports, Science, and Technology | | Kanji Furuya |
| Uehara Memorial Foundation | | Kanji Furuya |

| | | |
|---|---|---|
| Takeda Science Foundation | | Kanji Furuya |
| Cooperative Research Project Program of Joint Usage/Research Center at the Institute of Development, Aging and Cancer | | Kanji Furuya |
| Institute of Molecular Embryology and Genetics | | Kanji Furuya |
| KAKENHI (Grant-in-Aid for ScientificResearch) for Young Scientists | 24687023 | Kanji Furuya |
| KAKENHI Grants-in-Aid for Scientific Research on Innovative Areas | 23131515 | Kanji Furuya |
| KAKENHI Grants-in-Aid for Scientific Research on Innovative Areas | 25131707 | Kanji Furuya |
| KAKENHI | 17K07284 | Kanji Furuya |
| National Institute of Genetics | 2011-2017-A | Kanji Furuya |

The funders had no role in study design, data collection and interpretation, or the decision to submit the work for publication.

## Author contributions

Takeshi Wakida, Conceptualization, Data curation, Formal analysis, Investigation; Masae Ikura, Yoshiharu Shiroiwa, Toshiyuki Habu, Investigation, Methodology; Kenji Kuriya, Data curation, Investigation, Methodology; Shinji Ito, Resources, Data curation, Software, Formal analysis, Investigation, Methodology; Takuo Kawamoto, Supervision; Katsuzumi Okumura, Resources, Methodology; Tsuyoshi Ikura, Conceptualization, Supervision, Methodology, Writing—original draft, Writing—review and editing; Kanji Furuya, Conceptualization, Resources, Data curation, Software, Formal analysis, Supervision, Funding acquisition, Validation, Investigation, Visualization, Methodology, Writing—original draft, Project administration, Writing—review and editing

## Author ORCIDs

Kanji Furuya http://orcid.org/0000-0002-5099-8302

## Decision letter and Author response

Decision letter https://doi.org/10.7554/eLife.29953.028
Author response https://doi.org/10.7554/eLife.29953.029

# Additional files

## Supplementary files

• Supplementary file 1. Primers used in this study Primers used in this study was listed.
DOI: https://doi.org/10.7554/eLife.29953.023

• Supplementary file 2. Statistic data to construct *Figures 4B* and *6A,B* An excel formatted file was presented.
DOI: https://doi.org/10.7554/eLife.29953.024

• Supplementary file 3. The expanded photos of DNA fibers used in *Figure 5B* The field in white squares are region presented in *Figure 5B*.
DOI: https://doi.org/10.7554/eLife.29953.025

• Transparent reporting form
DOI: https://doi.org/10.7554/eLife.29953.026

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
