## [Decision Letter]

Thank you for submitting your article "CDK-PLK1 axis to target DNA checkpoint sensor protein RAD9 to tolerate genotoxic stress for promoting cell proliferation" for consideration by eLife. Your article has been reviewed by three peer reviewers, and the evaluation has been overseen by a Reviewing Editor and Jessica Tyler as the Senior Editor. The following individuals involved in review of your submission have agreed to reveal their identity: Dana Branzei (Reviewer #1); Rodrigo Bermejo (Reviewer #3).

The reviewers have discussed the reviews with one another and the Reviewing Editor has drafted this decision to help you prepare a revised submission.

Summary:

The 9-1-1 clamp plays a key role in activation of the DNA damage checkpoint. 9-1-1 is loaded at sites of DNA damage or replication-induced lesions, where it recruits the ATR kinase activator TOPBP1 for initiation of DNA damage checkpoint signaling. The recruitment of TOPBP1 to 9-1-1 is dependent on the CK2 kinase, which phosphorylates the 9-1-1 subunit RAD9 on serine 387 and serine 341. The role of these RAD9 phosphorylation sites has been well documented, however, much less is understood about additional phosphorylation events on RAD9, especially in mammalian systems. The manuscript by Furuya et al. reports that human RAD9 is also phosphorylated by the CDK and PLK1 kinases, and proposes that the coordinated action of CDK/PLK1 on RAD9 is important to counteract the DNA damage checkpoint. The authors claim that phosphorylation of RAD9 by CDK on threonine 292 creates a docking site for binding to PLK1, which in turn, further phosphorylates RAD9 on threonine 313 and serine 326. These claims are supported by in vitro and in vivo experiments. In addition, the authors generated human cell lines that stably express distinct Rad9 phospho-mutants and show that expression of these mutants results in enhanced checkpoint activation, reduced rates of DNA synthesis and slight sensitivity to hydroxyurea treatment.

Overall, all reviewers agreed that this is an interesting paper and that it provides important contributions to the field. The finding that the CDK and PLK1 kinases can modulate 9-1-1 function has important implications for understanding checkpoint control and the integration of checkpoint signaling with other cellular processes. The biochemical experiments were considered strong and well executed. Several useful reagents were generated, including phospho-specific antibodies and Rad9 point mutants.

Essential revisions:

1) All reviewers were concerned with the writing quality of the manuscript. A revised version of the manuscript will need to be much improved in this regard, and we strongly encourage editing by a native English speaker. The title of the manuscript also needs to be improved for clarity.

2) A major claim made by the authors is that CDK/PLK1-mediated RAD9 phosphorylation promote DNA synthesis, and they speculate that this happens largely by deregulating origin firing. However, as pointed out by the reviewers, the observed changes in DNA synthesis could be caused by changes in fork velocity. To discriminate between origin firing and fork speed, the authors should perform a molecular combing experiment in the presence of HU, where both fork rate and inter-origin distance can be measured. Genomic Vision provides such service in case the authors do not have the technique set up.

3) It is somewhat disappointing that the effects of expressing Rad9 phospho-mutants on checkpoint signaling, cell cycle and DNA synthesis are quite subtle. This could be caused by the presence of endogenous wild-type Rad9. Therefore, it would be worth generating a system to knockdown or knockout endogenous RAD9 in cell lines expressing the mutants.

4) The effect of expressing Rad9 phospho-mutants on cell proliferation was only examined in U2OS cells, which are rather unique in the sense that they use ALT for telomere lengthening and may have atypical mechanisms for checkpoint control. It is therefore important that the authors examine the effect of expressing Rad9 phospho-mutants in cell proliferation and DNA synthesis using other cells lines that do not use ALT.

5) The paper is mostly based on experiments using hydroxyurea-induced replication stress. The effect of other genotoxic stresses, such as DNA alkylation, doubled stranded breaks and inter-strand crosslinks should be tested. Examining how drugs such as MMS, CPT and MMC (and perhaps ionizing radiation) affect RAD9 phosphorylation by CDK and PLK1 will be useful to better characterize how these phosphorylation events are modulated. Furthermore, the analysis of DNA synthesis (EdU incorporation and FACS) in cells expressing Rad9 phospho-mutants treated with at least one of these other forms of genotoxicity will be useful.

6) The paper is lacking a better discussion on the mechanism by which RAD9 phosphorylation affect checkpoint signaling. More specifically, it would be useful if the authors expand on how Rad9 phosphorylation could be controlling its chromatin localization. For example, is phosphorylation preventing 9-1-1 loading? and promoting unloading from DNA?

7) The paper would also benefit from a more in-depth discussion of the physiological impact of Rad9 phosphorylation on the control of checkpoint response and cell proliferation. It is currently unclear if the authors are claiming that the action of CDK/PLK1 on RAD9 is part of a damage tolerance response (which promotes cell proliferation even in the presence of continued damage), or if it is a mechanism to promote quicker dampening of checkpoint signaling and timely resumption of cell cycle progression.

[Editors' note: further revisions were requested prior to acceptance, as described below.]

Thank you for resubmitting your work entitled "CDK-PLK1 axis targets DNA checkpoint sensor protein RAD9, promoting tolerance to genotoxic stress and cell proliferation" for further consideration at eLife. Your revised article has been favorably evaluated by Jessica Tyler (Senior editor), Marcus Smolka (Reviewing editor), and one of the reviewers.

The manuscript has been clearly improved on the experimental side. However, the written English was still considered poor. Therefore, you will need to significantly improve the text, perhaps by seeking help from a different company for improved writing services.

For the title, I would suggest the following:

"The CDK-PLK1 axis targets the DNA damage checkpoint sensor protein RAD9 to promote cell proliferation and tolerance to genotoxic stress"

Throughout the text, I urge you to refer to "DNA damage checkpoint" instead of "DNA checkpoint", since the latter is not very commonly used.

Please keep in mind that the written English of your whole manuscript needs to be significantly improved. For the abstract, I am pasting below a suggested version, which I have modified based on your latest abstract.

Suggested abstract:

"When proliferating cells encounter genotoxic stress, the DNA damage checkpoint is activated to assist DNA damage recovery by slowing down cell cycle progression. Thus, to drive proliferation, cells need to tolerate DNA damage and suppress the checkpoint response. However, the mechanism underlying this negative regulation of checkpoint activation is still elusive. Here we show that human Cyclin-Dependent-Kinases (CDKs) target the RAD9 subunit of the 9-1-1 checkpoint clamp on Thr292 for modulation of DNA damage checkpoint activation. Thr292 phosphorylation on RAD9 creates a binding site for Polo-Like-Kinase 1 (PLK1), which further phosphorylates RAD9 on Thr313. These CDK-PLK1-dependent phosphorylation events on RAD9 suppress checkpoint activation, therefore maintaining high rates of DNA synthesis in the presence of DNA replication stress. Our results suggest that CDK locally initiates a PLK1-dependent signaling response that antagonizes the ability of the DNA damage checkpoint to detect DNA damage. These findings provide a mechanism for how replicating cells suppress DNA damage checkpoint signaling to promote proliferation in the presence of genotoxic stress."

---

## [Author Response]

Essential revisions:

1) All reviewers were concerned with the writing quality of the manuscript. A revised version of the manuscript will need to be much improved in this regard, and we strongly encourage editing by a native English speaker. The title of the manuscript also needs to be improved for clarity.

Our manuscript was checked (grammar) by a professional company.

We also changed the title from “PLK1-CDK axis to target DNA checkpoint sensor protein RAD9 to tolerate genotoxic stress for promoting cell proliferation” to “CDK-PLK1 axis targets DNA checkpoint sensor protein RAD9, promoting tolerance to genotoxic stress and cell proliferation”.

2) A major claim made by the authors is that CDK/PLK1-mediated RAD9 phosphorylation promote DNA synthesis, and they speculate that this happens largely by deregulating origin firing. However, as pointed out by the reviewers, the observed changes in DNA synthesis could be caused by changes in fork velocity. To discriminate between origin firing and fork speed, the authors should perform a molecular combing experiment in the presence of HU, where both fork rate and inter-origin distance can be measured. Genomic Vision provides such service in case the authors do not have the technique set up.

Thank you for this comment. We performed DNA combing assay to look into a molecular level to assess which steps of DNA checkpoint dependent processes are affected in the RAD9 mutant (RAD9-T313A; RAD9 that is not phosphorylated by PLK1). The results (Figure 5) gave us rather unexpected observation, especially in non-stressed cells, but it fits our model. First, when cells were under replicative stresses (HU), the replication fork rates were suppressed to the same level both in WT and T313A mutant. This indicates that the reduction observed in the bulk DNA synthesis rate in T313A mutant was not due to the reduction in replication fork rate. Furthermore, we observed a slight increase in inter-origin distance, which suggests that the dormant origins were rather repressed, in RAD9-T313A mutant cells. Although the difference is marginally significant (IOD; WT: 61 ± 38 kb, T313A: 69 ± 41 kb, p-value; 0.058), it is possible that the repression of dormant origin firing could be the part of the reason for reduction in bulk DNA synthesis rate in T313A mutant.

The finding we had in untreated cells was unexpected. When the mutant cells were not under the stress, we observed a great reduction in replication fork rate in T313A mutant (WT: 0.85 ± 0.36 kb/minutes, T313A: 0.66 ± 0.27 kb/minutes, p-value: 5.8 x10-7). In contrast, the inter-origin distances (IOD) were shortened in T313A mutant (WT: 131 ± 62 kb, T313A: 107 ± 61 kb, p-value: 8.0 x 10-8). This reduction in IOD suggested more activation of dormant origins compared to the WT and could indicate that reduction in the fork rate can be compensated by increasing efficiency of firing (dormant) origins, as suggested in some reports (e.g.: Iyer and Rhind, 2013). These combing data were obtained from Genomic Vision.

In parallel, we also examined the fork rate under aphidicolin-induced stress (Figure 5—figure supplement 1) as we had already observed a slight reduction in bulk DNA synthesis rate in aphidicolin-treated cells (Figure 4—figure supplement 6). The experiments were performed by one of our authors. We saw the fork rate was also similarly suppressed both in WT and T313A cells under aphidicolin-induced stress and that again suggested that the reduction in bulk DNA synthesis rate was not due to reduction in the fork rate. [We should note that there are technical differences between Genomic Vision and our work, as absolute values of the fork rate we obtained was smaller than the value of that we obtained from Genomic Vision. This is partly caused by the different buffer, glasses or technique used upon stretching DNA fiber on the glass, between us and Genomic Vision. Furthermore, difference in evaluators for the DNA forks may pick the track in different criteria (cut off size of the minimal fork size, how to exclude the fusion forks, etc). We alluded to this issue in the text and in the Materials and methods section].

We interpreted the above data in a non-stressed condition – in HU and in aphidicolin – as follows: PLK1-dependent phosphorylation on RAD9 is functioning even during normal S phase (that supports why T292/T313 phosphorylation could be seen even in non-stressed cells), and defective phosphorylation may lead to the moderate activation of S phase checkpoint that could reduce the fork speed, and in those situation, possibly dormant (or other) origin fires, and the cells compensate bulk DNA synthesis rate. However, when they suffer from 0.2 mM HU, although WT cells somehow drive cell proliferation possibly by activating dormant or late origin firing, T313A mutants failed to proliferate because the excess level of S-phase checkpoint activation may inhibit the origin firings that should help maintaining rate of bulk DNA synthesis. Thus, we think that controlling RAD9 is important to have an appropriate level of checkpoint activation to proliferate under the genotoxic stress.

We added Figure 5 (HU 0.2 mM) and Figure 5—figure supplement 1 (Aphidicolin) to show this point. Also, because these data somewhat responded to our previous discussion, we rearranged the flow of the sections within the Discussion (although the order of the Discussion section has been rearranged, the content is unchanged). We believe that the present form of the Discussion is more fluent and easier to follow. The next text in the Results, Discussion Materials and methods sections follows:

Subsection “PLK1-dependent phosphorylation of RAD9 Thr313 is required to suppress S-phase checkpoint response”:

“We went on to perform DNA combing assay to see which steps of DNA replication contribute to slow down bulk DNA synthesis. […] Also, it should be noted that the replication fork rate was already slower in RAD9-T313A expressing cells from non-stressed condition, which may suggest the enhanced checkpoint response in RAD9-T313A expressing cells in normal cell cycle (see Discussion section).”

Subsection “Phosphorylation of the checkpoint sensor protein by CDK and PLK1 is required for maintaining the rate of DNA replication upon genotoxic stress”:

“When we performed DNA fiber/combing assay in HU treated condition, we observed that velocity of replication forks was the same in WT- and T313A-RAD9 expressing cells. […] In T313A-expressing cells, further encounter to HU-induced stress would activate excess DNA damage response and could cause a reduction in number of firing origins, which would result in slower rate of S-phase progression.”

Subsection “DNA molecular combing assay”:

DNA combing assay was performed as described (Conti et al., 2007; Kuriya et al., 2015; Michalet et al., 1997). […] Thus, we emphasize on the relative difference in values within the samples obtained from the same series.”

3) It is somewhat disappointing that the effects of expressing Rad9 phospho-mutants on checkpoint signaling, cell cycle and DNA synthesis are quite subtle. This could be caused by the presence of endogenous wild-type Rad9. Therefore, it would be worth generating a system to knockdown or knockout endogenous RAD9 in cell lines expressing the mutants.

Thank you for the comment. Our original RAD9 plasmids used to create stable cell line were actually constructed as siRNA resistant form, and we knocked down endogenous RAD9 to see whether the effect of phosphorylation defective mutants (see Figure 4—figure supplement 4). We see some difference compared to cells that did not apply siRAD9. As for the T313A-mutated RAD9, we see the enhancement of checkpoint phenotype (CHK1-pS345, Figure 4—figure supplement 4). We also see a slight reduction of DNA replication rate assessed by EdU incorporation compared to the control cells (Figure 4—figure supplement 4). As for S326A-mutated RAD9 expressing cells that did not show the checkpoint phenotype as we reported in this manuscript, we did not see neither increase in CHK1-pS345 phosphorylation nor decrease in DNA replication rate compared to siRNA-control treated cells (Figure 4—figure supplement 4). And we concluded that S326A-mutated RAD9 cells do not show accelerated checkpoint activation phenotype. Thus, we added the text as follows:

Subsection “PLK1-dependent phosphorylation of RAD9 Thr313 is required to suppress S-phase checkpoint response”:

“We further knocked down endogenous RAD9 (Figure 4—figure supplement 4) to see whether the observed cellular effect by the expression of T313A- or S326A-mutated RAD9 is enhanced. […] In cells that expressed S326A-mutated RAD9, silencing did not result in an obvious enhancement in phosphorylation of Ser345 of CHK1 or in a decrease in DNA replication rate (Figure 4—figure supplement 4).”

4) The effect of expressing Rad9 phospho-mutants on cell proliferation was only examined in U2OS cells, which are rather unique in the sense that they use ALT for telomere lengthening and may have atypical mechanisms for checkpoint control. It is therefore important that the authors examine the effect of expressing Rad9 phospho-mutants in cell proliferation and DNA synthesis using other cells lines that do not use ALT.

Thank you for the comment; this was very helpful. We tested in HEK293A cells that was reportedly telomerase-positive (Bryan et al., 1995). As for the checkpoint phenotype using HEK293A, we have already performed in Figure 1 and Figure 4, and showed increased chromatin-bound CHK1-pS345. Here, we have added the data to show the reduction in the rate of DNA replication could be observed in HEK293A cells in Figure 4—figure supplement 5. We also performed proliferation assay and showed that RAD9-T292A and RAD9-T313A both presented decreased ability to proliferate under HU stress (Figure 6—figure supplement 1). We added the text as follows:

Subsection “PLK1-dependent phosphorylation of RAD9 Thr313 is required to suppress S-phase checkpoint response”:

“Furthermore, we also confirmed the reduction in DNA replication rate in telomerase-positive HEK293A cells (Bryan et al., 1995) when either T292A- or T313A-mutated RAD9 was expressed (Figure 4—figure supplement 5). This result excluded the possibility that the slowing of DNA replication rate was caused by the atypical mechanism for checkpoint control which may occur in U2OS cells via Alternative Lengthening of Telomeres (ALT) (Figure 4—figure supplement 5).”

Subsection “CDK- and PLK1- dependent phosphorylation of RAD9 allows cells to grow under genotoxic stress”:

“Indeed, we observed that cells expressing T292A- or T313A-mutated RAD9 exhibited a slower growth rate in the presence of HU (As for U2OS cells: Figure 6, As for HEK293A cells: Figure 6—figure supplement 1).”

5) The paper is mostly based on experiments using hydroxyurea-induced replication stress. The effect of other genotoxic stresses, such as DNA alkylation, doubled stranded breaks and inter-strand crosslinks should be tested. Examining how drugs such as MMS, CPT and MMC (and perhaps ionizing radiation) affect RAD9 phosphorylation by CDK and PLK1 will be useful to better characterize how these phosphorylation events are modulated. Furthermore, the analysis of DNA synthesis (EdU incorporation and FACS) in cells expressing Rad9 phospho-mutants treated with at least one of these other forms of genotoxicity will be useful.

Thank you for the comment. We have tested to see the phosphorylation status in response to MMS, MMC, CPT, IR or Aphidicolin (Figure 3—figure supplement 3). Since we only see a little checkpoint response in MMS (0.001%) and MMC (0.2ug/ml), we re-did the experiment using higher doses (Figure 3—figure supplement 3). The drugs were added in average one hour before harvest, so as not to disturb the overall cell cycle profiles. In most cases, we saw either sustained or a slight increased efficiency in PLK1-dependent phosphorylation: pT313 or pS326.

We have further tested two of these stresses (MMC and aphidicolin, Figure 4—figure supplement 6) for the EdU analysis to measure the rate of bulk DNA replication. In aphidicolin-treated cells, we observed reduction in DNA replication rate to 90% in RAD9-T313A mutant cells. In MMC-treated cells, although we did not observe the obvious reduction in the rate of DNA replication, we observed more cells accumulated in early S phase in RAD9-T313A mutant cells.

These indicated that, as for the phenotype of reduction in DNA replication rate, it could be specific to the direct DNA replication inhibitor, but not to the DNA damaging agents. Thus, we added text to explain the results for the phosphorylation status as follows (subsection “PLK1 phosphorylates Thr313 and Ser326 on RAD9 in vivo”), and the EdU assay with aphidicolin or MMC-treated cells in (subsection “PLK1-dependent phosphorylation of RAD9 Thr313 is required to suppress S-phase checkpoint response”);

Subsection “PLK1 phosphorylates Thr313 and Ser326 on RAD9 in vivo”:

“We further tested the effects of other types of DNA damage on PLK1-dependent phosphorylation of RAD9. Treatment with camptothecin (CPT: topoI inhibitor), ionizing radiation (IR), or high doses of methylmethanesulfonate (MMS: 0.03%) or mitomycin C (MMC:1 μg/ml) yielded slight increases in pSer326 phosphorylation (Figure 3—figure supplement 3). Aphidicolin (Aph: a DNA polymerase inhibitor) treatment reduced the levels of both pThr313 and pSer 326, however, pThr292 was also decreased, and thus these results suggested that the efficiency of PLK1-dependent phosphorylation is either maintained or slightly enhanced in response to most types of genotoxic stresses (Figure 3—figure supplement 5).”

Subsection “PLK1-dependent phosphorylation of RAD9 Thr313 is required to suppress S-phase checkpoint response”:

“A slowing of DNA replication rate also was observed in response to other types of DNA damage. Specifically, aphidicolin treatment also reduced the rate of EdU incorporation, and MMC treatment caused more cells to accumulate in early S phase when T313A-mutated RAD9 was expressed, although the latter treatment did not yield an obvious decrease in EdU incorporation (Figure 4—figure supplement 6). The observed phenotype may have been specific to the direct inhibition of DNA replication by some agents (HU, a dNTP synthesis inhibitor; aphidicolin, a DNA polymerase inhibitor) but not by others (MMC, a DNA damaging agent that crosslinks DNA strands).”

6) The paper is lacking a better discussion on the mechanism by which RAD9 phosphorylation affect checkpoint signaling. More specifically, it would be useful if the authors expand on how Rad9 phosphorylation could be controlling its chromatin localization. For example, is phosphorylation preventing 9-1-1 loading? and promoting unloading from DNA?

Thank you for the comment. It is quite difficult to distinguish whether the chromatin localization of RAD9 is controlled via loading or unloading. At present how 9-1-1 is unloaded from DNA damage sites is not fully understood. It could be possible that the clamp loader RAD17 could act as the unloader of 9-1-1. We think that, in our case, whether 9-1-1 is loaded or unloaded on chromatin could depend on the affinity of the 9-1-1 complex itself to DNA damage site (e.g. whether RPA-binding motif on RAD9 is available or not). At least we think the phospho-dependent chromatin dissociation is not controlled via regulating the interaction to a clamp loader RAD17, as T292A mutation did not affect the binding to it (IP analysis; data not shown).

We added the text as follows (subsection “CDK and PLK1 target RAD9 to modulate DNA damage detection”:

“The sites of PLK1-dependent phosphorylation of RAD9 are located within the tail region. This region of RAD9 is expected to help the protein’s association to DNA damage sites, for example the region phosphorylate by PLK1 is close to the region required for association to the single-stranded-DNA binding protein RP-A (Figure 3—figure supplement 1) (Xu et al., 2008). Furthermore, we did not observe a decrease in the amount of the RAD9-bound clamp-loader RAD17 in co-IP experiment when T292 was mutated, even under condition of genotoxic stress (data not shown). Thus, PLK1-dependent phosphorylation may affect the affinity of the 9-1-1 complex itself to the DNA damage sites.”

7) The paper would also benefit from a more in-depth discussion of the physiological impact of Rad9 phosphorylation on the control of checkpoint response and cell proliferation. It is currently unclear if the authors are claiming that the action of CDK/PLK1 on RAD9 is part of a damage tolerance response (which promotes cell proliferation even in the presence of continued damage), or if it is a mechanism to promote quicker dampening of checkpoint signaling and timely resumption of cell cycle progression.

Thank you for the comment. Since we are presenting the data to show the increased amount of RAD9 onto the chromatin in the mutant, we propose that PLK1 antagonize the function of DNA damage sensing ability of RAD9. Thus, we think it controls at the activation step of checkpoint rather than the dampening the activated checkpoint. And, at the moment, we think this is a part of the novel form of DNA tolerance mechanism. Our results may explain why PLK1 overexpression is linked to cancer cells as we discussed in the last paragraph of the Discussion: “In this scenario, PLK1 may raise a threshold of DNA damage levels, which would require more DNA damage stress for the S phase checkpoint activation. In other words, PLK1-dependent phosphorylation of RAD9 may suppress the activation step of the checkpoint, rather than dampening the activated DNA checkpoint processes, and this process could act as a novel form of DNA damage tolerance mechanism to facilitate cell proliferation.”

[Editors' note: further revisions were requested prior to acceptance, as described below.]

Thank you for resubmitting your work entitled "CDK-PLK1 axis targets DNA checkpoint sensor protein RAD9, promoting tolerance to genotoxic stress and cell proliferation" for further consideration at eLife. Your revised article has been favorably evaluated by Jessica Tyler (Senior editor), Marcus Smolka (Reviewing editor), and one of the reviewers.

The manuscript has been clearly improved on the experimental side. However, the written English was still considered poor. Therefore, you will need to significantly improve the text, perhaps by seeking help from a different company for improved writing services.

Thank you for the suggestion. We sought English editing services from a native English speaker (a recognized scientific English editor), and we believe that the manuscript is now easier to read. We also changed the term “DNA checkpoint” to “DNA damage checkpoint”. The content of the manuscript remains unchanged. However, upon the initial checking process, a member of the eLife staff has suggested to re-assemble the figures, and we have swapped the order of “Figure 3—figure supplement 3” and “Figure 3—figure supplement 3”. Accordingly, the text has been modified.

For the title, I would suggest the following:

"The CDK-PLK1 axis targets the DNA damage checkpoint sensor protein RAD9 to promote cell proliferation and tolerance to genotoxic stress"

Throughout the text, I urge you to refer to "DNA damage checkpoint" instead of "DNA checkpoint", since the latter is not very commonly used.

Please keep in mind that the written English of your whole manuscript needs to be significantly improved. For the abstract, I am pasting below a suggested version, which I have modified based on your latest abstract.

Suggested abstract:

"When proliferating cells encounter genotoxic stress, the DNA damage checkpoint is activated to assist DNA damage recovery by slowing down cell cycle progression. Thus, to drive proliferation, cells need to tolerate DNA damage and suppress the checkpoint response. However, the mechanism underlying this negative regulation of checkpoint activation is still elusive. Here we show that human Cyclin-Dependent-Kinases (CDKs) target the RAD9 subunit of the 9-1-1 checkpoint clamp on Thr292 for modulation of DNA damage checkpoint activation. Thr292 phosphorylation on RAD9 creates a binding site for Polo-Like-Kinase 1 (PLK1), which further phosphorylates RAD9 on Thr313. These CDK-PLK1-dependent phosphorylation events on RAD9 suppress checkpoint activation, therefore maintaining high rates of DNA synthesis in the presence of DNA replication stress. Our results suggest that CDK locally initiates a PLK1-dependent signaling response that antagonizes the ability of the DNA damage checkpoint to detect DNA damage. These findings provide a mechanism for how replicating cells suppress DNA damage checkpoint signaling to promote proliferation in the presence of genotoxic stress."

The abstract has been modified. We adopted the abstract suggested in the previous letter. However, since the number of words exceeded 150 words, we asked the English editor to shorten it. Thus, the included abstract is now as follows:

“Genotoxic stress causes proliferating cells to activate the DNA damage checkpoint, to assist DNA damage recovery by slowing cell cycle progression. Thus, to drive proliferation, cells must tolerate DNA damage and suppress the checkpoint response. However, the mechanism underlying this negative regulation of checkpoint activation is still elusive. We show that human Cyclin-Dependent-Kinases (CDKs) target the RAD9 subunit of the 9-1-1 checkpoint clamp on Thr292, to modulate DNA damage checkpoint activation. Thr292 phosphorylation on RAD9 creates a binding site for Polo-Like-Kinase1 (PLK1), which phosphorylates RAD9 on Thr313. These CDK-PLK1-dependent phosphorylations of RAD9 suppress checkpoint activation, therefore maintaining high DNA synthesis rates during DNA replication stress. Our results suggest that CDK locally initiates a PLK1-dependent signaling response that antagonizes the ability of the DNA damage checkpoint to detect DNA damage. These findings provide a mechanism for the suppression of DNA damage checkpoint signaling, to promote cell proliferation under genotoxic stress conditions.”

We also made the following changes:

Change: “DNA checkpoint” => “DNA damage checkpoint” throughout the manuscript.

Change: “DNA checkpoint” => “DNA damage checkpoint in S phase” in the Introduction.